# LIMIS: Locally Interpretable Modeling using Instance-wise Subsampling

**Jinsung Yoon, Sercan Ö. Arik, Tomas Pfister**     *{jinsungyoon, soarik, tpfister}@google.com*
*Google Cloud AI*

**Reviewed on OpenReview:** *https://openreview.net/forum?id=S8eABAy8P3*

## Abstract

Understanding black-box machine learning models is crucial for their widespread adoption. Learning globally interpretable models is one approach, but achieving high performance with them is challenging. An alternative approach is to explain individual predictions using locally interpretable models. For locally interpretable modeling, various methods have been proposed and indeed commonly used, but they suffer from low fidelity, i.e. their explanations do not approximate the predictions well. In this paper, our goal is to push the state-of-the-art in high-fidelity locally interpretable modeling. We propose a novel framework, Locally Interpretable Modeling using Instance-wise Subsampling (LIMIS). LIMIS utilizes a policy gradient to select a small number of instances and distills the black-box model into a low-capacity locally interpretable model using those selected instances. Training is guided with a reward obtained directly by measuring the fidelity of the locally interpretable models. We show on multiple tabular datasets that LIMIS near-matches the prediction accuracy of black-box models, significantly outperforming state-of-the-art locally interpretable models in terms of fidelity and prediction accuracy.

## 1 Introduction

In many real-world applications, machine learning is required to be interpretable – doctors need to understand why a particular treatment is recommended, banks need to understand why a loan is declined, and regulators need to investigate systems against potential fallacies (Rudin, 2018). On the other hand, the machine learning models that have made the most significant impact via predictive accuracy improvements, such as deep neural networks (DNNs) and ensemble decision tree (DT) variants (Goodfellow et al., 2016; He et al., 2016; Chen & Guestrin, 2016; Ke et al., 2017), are 'black-box' in nature – their decision making is based on complex non-linear interactions between many parameters that are difficult to interpret. Many studies have suggested a trade-off between performance and interpretability (Virág & Nyitrai, 2014; Johansson et al., 2011; Lipton, 2016). While globally interpretable models such as linear models or shallow Decision Trees (DTs) have simple explanations for the entire model behaviors, they generally yield significantly worse performance than black-box models.

One alternative approach is locally interpretable modeling – explaining a single prediction individually instead of explaining the entire model (Ribeiro et al., 2016). A globally interpretable model fits a single interpretable model to the entire data, while a locally interpretable model fits an interpretable model locally, i.e. for each instance/sample individually, by distilling knowledge from a black-box model around the observed sample. Locally interpretable models are useful for real-world AI deployments by providing succinct and human-like explanations via locally fitted models. They can be utilized to identify systematic failure cases (e.g. by seeking common trends in how failure cases depend on the inputs) (Mangalathu et al., 2020), detect biases (e.g. by quantifying the importance of a particular feature) (ElShawi et al., 2021), provide actionable feedback to improve a model (e.g. suggesting what training data to collect) (Ribeiro et al., 2016), and for counterfactual analyses (e.g. by investigating the local model behavior around the observed data sample) (Grath et al., 2018).

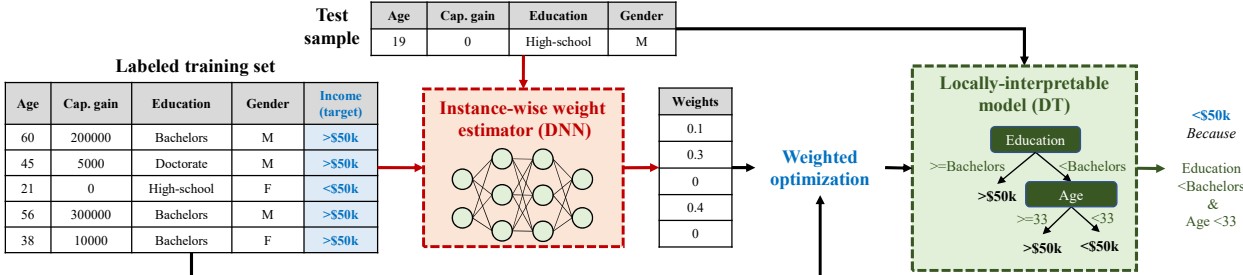

Figure 1: LIMIS example for the income classification task. For each test sample, the most valuable training samples are chosen to fit the locally-interpretable model (DT here), and it provides human-like explanations to the decision. More use-cases for human-in-the-loop AI capabilities of LIMIS can be found in Sect. 6

Various methods have been proposed for locally interpretable modeling: Local Interpretable Model-agnostic Explanations (LIME) (Ribeiro et al., 2016), Supervised Local modeling methods (SILO) (Bloniarz et al., 2016), and Model Agnostic Supervised Local Explanations (MAPLE) (Plumb et al., 2018). LIME in particular has gained significant popularity. Yet, the locally interpretable modeling problem is still far from as being solved. To be useful in practice, a locally interpretable model should have high fidelity, i.e, it should approximate the 'black-box' model well (Plumb et al., 2019; Lakkaraju et al., 2019). Recent studies have shown that LIME indeed often yields low fidelity (Alvarez-Melis & Jaakkola, 2018; Zhang et al., 2019; Ribeiro et al., 2018; Lakkaraju et al., 2017); indeed, as we show in Sec. 5, in some cases, LIME's performance is even worse than simple globally interpretable models. The performance of other methods such as SILO and MAPLE are also far from the achievable limits. Overall, locally interpretable modeling while ensuring high fidelity across a wide range of cases is an everlasting challenging problem, and we propose that it requires a substantially-novel design for the fitting paradigm. A fundamental challenge to fit a locally interpretable model is the representational capacity difference when applying distillation. Black-box models, such as DNNs or ensemble DTs, have much larger capacity compared to interpretable models. This can result in underfitting with conventional distillation techniques and consequently suboptimal performance (Hinton et al., 2015; Wang et al., 2019).

To address the fundamental challenges aforementioned above, we propose a novel instance-wise subsampling method to fit Locally Interpretable Models, named *LIMIS*, that is motivated by meta-learning (Ren et al., 2018). Fig. 1 depicts LIMIS for the income classification task. LIMIS utilizes the instance-wise weight estimator to identify the importance of the training samples to explain the test sample. Then, it trains a locally-interpretable model with weighted optimization to return the accurate prediction and corresponding local explanations. LIMIS efficiently tackles the distillation challenge by fitting the locally interpretable model with a small number of instances/samples that are determined to be most valuable to maximize the fidelity. Unlike alternative methods that apply some supervised learning approaches to determine valuable instances, LIMIS learns an instance-wise weight estimator (modeled with a DNN) directly using the fidelity metric for selection. Accurate determination of the most valuable instances allows the locally interpretable model to more effectively utilize its small representational capacity. At various regression and classification tasks, we demonstrate that LIMIS significantly outperforms alternatives. In most cases, the locally *interpretable* models obtained by LIMIS near-match the performance of the complex black-box models that they are trained to interpret. In addition, LIMIS offers the instance-based explainability via ranking of the most valuable training instances. We also show that the high-fidelity explanations can open new horizons for reliable counterfactual analysis, by understanding what input modification would change the outcome, which can be important for human-in-the-loop AI deployments (see Sec. 6.2).

## 2    Related Work

**Locally interpretable models:** There are various approaches to interpret black-box models (Gilpin et al., 2018). One is to directly decompose the prediction into feature attributions, e.g. Shapley values (Štrumbelj & Kononenko, 2014) and their computationally-efficient variants (Lundberg & Lee, 2017). Others are based on

activation differences, e.g. DeepLIFT (Shrikumar et al., 2017), or saliency maps using the gradient flows, e.g. CAM (Zhou et al., 2016) and Grad-CAM (Selvaraju et al., 2017). In this paper, we focus on the direction of locally interpretable modeling – distilling a black-box model into an interpretable model for each instance in tabular domains. LIME (Ribeiro et al., 2016) is the most commonly used method for locally interpretable modeling in tabular domains. LIME is based on modifying the input feature values and learning from the impact of the modifications on the output. A fundamental challenge for LIME is the meaningful distance metric to determine neighborhoods, as simple metrics like Euclidean distance may yield poor fidelity, and the estimation is highly sensitive to normalization (Alvarez-Melis & Jaakkola, 2018). SILO (Bloniarz et al., 2016)) proposed a nonparametric regression based on fitting small-scale local models which can be utilized for locally interpretable models similar to LIME. It determines the neighborhoods for each instance using tree-based ensembles – it utilizes DT ensembles to determine the weights of training instances for each test instance and uses the weights to optimize a locally interpretable model. Note that these weights are independent of the locally interpretable models. MAPLE (Plumb et al., 2018) further adds feature selection on top of SILO. SILO and MAPLE optimize the DT-based ensemble methods independently and this disjoint optimization results in suboptimal performance. To fit a proper locally interpretable model, a key problem is the selection of the appropriate training instances for each test instance. LIME uses Euclidean distances, whereas SILO and MAPLE use DT-based ensemble methods. Our proposed method, LIMIS, takes a very different approach: to efficiently explore the large search space, we directly optimize the instance-wise subsampler with the fidelity as the reward.

**Data-weighted training:** Optimal weighting of training instances is a paramount problem in machine learning. By upweighting/downweighting the high/low value instances, better performance can be obtained in certain scenarios, such as with noisy labels (Jiang et al., 2018). One approach for data weighting is utilizing influence functions (Koh & Liang, 2017), that are based on oracle access to gradients and Hessian-vector products. Jointly-trained student-teacher method constitutes another approach (Jiang et al., 2018; Bengio et al., 2009) to learn a data-driven curriculum. Using the feedback from the teacher, instance-wise weights are learned by the student model. Aligned with our motivations, meta learning is considered for data weighting in Ren et al. (2018). Their proposed method utilizes gradient descent-based meta learning, guided by a small validation set, to maximize the target performance. LIMIS utilizes data-weighted training for a novel goal: interpretability. Unlike gradient descent-based meta learning, LIMIS uses policy gradient and integrates the fidelity metric as the reward. Aforementioned works (Jiang et al., 2018; Koh & Liang, 2017; Bengio et al., 2009; Ren et al., 2018) estimate the same ranking of training data for all instances. Instead, LIMIS yields an instance-wise ranking of training data, enabling efficient distillation of a black-box model prediction into a locally interpretable model. Yeh et al. (2018) can also provide instance-wise ranking of training samples but for sample-based explainability. Differently, LIMIS utilizes instance-wise ranking with the objective of locally-interpretable modeling.

## 3 LIMIS Framework

Consider a training dataset $\mathcal{D} = \{(\mathbf{x}_i, y_i)\}_{i=1}^{N} \sim \mathcal{P}$ for a black-box model $f$, where $\mathbf{x}_i \in \mathcal{X}$ are $d$-dimensional feature vectors and $y_i \in \mathcal{Y}$ are the corresponding labels. We also assume a probe dataset $\mathcal{D}^p = \{(\mathbf{x}_j^p, y_j^p)\}_{j=1}^{M} \sim \mathcal{P}$, to evaluate the model performance to guide meta-learning as in Ren et al. (2018). If there is no explicit probe dataset, it can be randomly split from the training dataset ($\mathcal{D}$).

### 3.1 Training and inference

LIMIS is composed of: **(i) Black-box model** $f : \mathcal{X} \to \mathcal{Y}$ – any machine learning model to be explained (e.g. a DNN), **(ii) Locally interpretable model** $g_\theta : \mathcal{X} \to \mathcal{Y}$ – an inherently-interpretable model by design (e.g. a shallow DT), **(iii) Instance-wise weight estimation model** $h_\phi : \mathcal{X} \times \mathcal{X} \times \mathcal{Y} \to [0, 1]$ – a function that outputs the instance-wise weights to fit the locally interpretable model, specifying for each instance how valuable it is for training the locally interpretable model. It takes its input as the concatenation of a probe instance's feature, a training instance's feature, and a corresponding black-box model prediction. It can be a complex ML model – here a DNN.

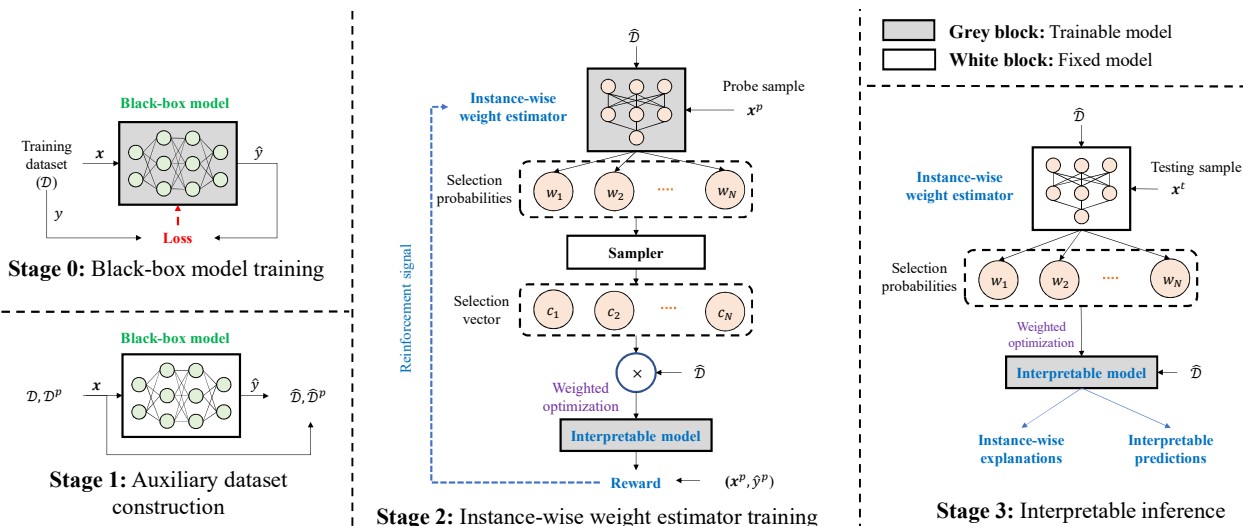

Figure 2: Block diagram of the proposed method. White blocks represent fixed (not learnable) models, and grey blocks represent trainable (learnable) models. **Stage 0:** Black-box model training. **Stage 1:** Auxiliary dataset construction. **Stage 2:** Instance-wise weight estimator training. **Stage 3:** Interpretable inference.

Our goal is to construct an accurate locally interpretable model $g_\theta$ such that the prediction made by it is similar to the prediction of the trained black-box model $f^*$ – i.e. the locally interpretable model has high fidelity. We use a loss function, $\mathcal{L} : \mathcal{Y} \times \mathcal{Y} \to \mathbb{R}$ to quantify the fidelity of the locally interpretable model which measures the prediction differences between black-box model and locally interpretable model (e.g. in terms of mean absolute error).

The locally interpretable model has a significantly lower representational capacity compared to the black-box model. This is the bottleneck that LIMIS aims to address. Ideally, to avoid underfitting, such low-capacity interpretable models should be learned with a minimal number of training instances that are most effective in capturing the model behavior. We propose an instance-wise weight estimation model $h_\phi$ to output the likelihood of each training instance being used for fitting the locally interpretable model. Integrating this with the goal of training an accurate locally interpretable model yields the following objective:

$$\min_{h_\phi} \quad \mathbb{E}_{\mathbf{x}^p \sim P_X} \left[ \mathcal{L}(f^*(\mathbf{x}^p), g^*_{\theta(\mathbf{x}^p)}(\mathbf{x}^p)) \right] + \lambda \mathbb{E}_{\mathbf{x}^p, \mathbf{x} \sim P_X} \left[ h_\phi(\mathbf{x}^p, \mathbf{x}, f^*(\mathbf{x})) \right]$$
$$\text{s.t.} \quad g^*_{\theta(\mathbf{x}^p)} = \arg\min_{g_\theta} \mathbb{E}_{\mathbf{x} \sim P_X} \left[ h_\phi(\mathbf{x}^p, \mathbf{x}, f^*(\mathbf{x})) \cdot \mathcal{L}_g(f^*(\mathbf{x}), g_\theta(\mathbf{x})) \right], \tag{1}$$

where $\lambda \geq 0$ is a hyper-parameter to control the number of training instances used to fit the locally interpretable model, and $h_\phi(\mathbf{x}^p, \mathbf{x}, f^*(\mathbf{x}))$ is the weight for each training pair $(\mathbf{x}, f^*(\mathbf{x}))$ and for the probe data $\mathbf{x}^p$. $\mathcal{L}_g$ is the loss function to fit the locally interpretable model (here to minimize the mean squared error) between the predicted values for regression and logits for classification. $\phi$ and $\theta$ are the trainable parameters, whereas $f^*$ (the pre-trained black-box model) is fixed. The first term in the objective function $\mathbb{E}_{\mathbf{x}^p \sim P_X} \left[ \mathcal{L}(f^*(\mathbf{x}^p), g^*_{\theta(\mathbf{x}^p)}(\mathbf{x}^p)) \right]$ is the fidelity metric, representing the prediction differences between the black-box model and locally interpretable models. The second term in the objective function $\mathbb{E}_{\mathbf{x}^p, \mathbf{x} \sim P_X} \left[ h_\phi(\mathbf{x}^p, \mathbf{x}, f^*(\mathbf{x})) \right]$ represents the expected number of selected training points to fit the locally interpretable model. Lastly, the constraint ensures that the locally interpretable model is derived from weighted optimization, where weights are the outputs of $h_\phi$. Our formulation does not assume any constraint on $g_\theta$ – it can be any inherently interpretable model. In experiments, we use simple decision tree or regression model (with closed-form solution) so that the complexity of the constraint optimization is negligible. Note that we utilize a deep model for weight optimization ($h_\phi$) but a simple interpretable model for explanation ($g_\theta$). LIMIS encompasses 4 stages:

● **Stage 0 − Black-box model training**: Given the training set $\mathcal{D}$, the black-box model $f$ is trained to minimize a loss function $\mathcal{L}_f$ (e.g. mean squared error for regression or cross-entropy for classification), i.e.,

$f^* = \arg\min_f \frac{1}{N} \sum_{i=1}^{N} \mathcal{L}_f(f(\mathbf{x}_i), y_i)$. If there exists a pre-trained black-box model, we can skip this stage and retrieve the given pre-trained model as $f^*$.

• **Stage 1** – **Auxiliary dataset construction**: Using the pre-trained black-box model $f^*$, we create auxiliary training and probe datasets, as $\hat{\mathcal{D}} = \{(\mathbf{x}_i, \hat{y}_i), i = 1, ..., N\}$ (where $\hat{y}_i = f^*(\mathbf{x}_i)$) and $\hat{\mathcal{D}}^p = \{(\mathbf{x}_j^p, \hat{y}_j^p), j = 1, ..., M\}$ (where $\hat{y}_j^p = f^*(\mathbf{x}_j^p)$), respectively. These auxiliary datasets ($\hat{\mathcal{D}}, \hat{\mathcal{D}}^p$) are used for training the instance-wise weight estimation model and locally interpretable model.

If we want to understand the black-box models that are trained on the given datasets, Stage 0 and Stage 1 are necessary because the main objective is to understand the local dynamics of the black-box models' decision boundary. On the other hand, if we want to understand the local dynamics of the given datasets, we can skip Stage 0 and 1 and directly utilize the given dataset for Stage 2 and 3.

• **Stage 2** – **Instance-wise weight estimator training**: LIMIS employs an instance-wise weight estimator to output selection probabilities that yield the selection weights, and the selection weights determine the fitted local interpretable model via weighted optimization. We train the instance-wise weight estimator using the auxiliary datasets ($\hat{\mathcal{D}}, \hat{\mathcal{D}}^p$). The search space for all sample weights would be very large, and for efficient search, proper exploration is crucial. To this end, we consider probabilistic selection with a sampler block that is based on the output of the instance-wise weight estimator – $h_\phi(\mathbf{x}_j^p, \mathbf{x}_i, \hat{y}_i)$ represents the probability that $(\mathbf{x}_i, \hat{y}_i)$ is selected to train a locally interpretable model for the probe instance $\mathbf{x}_j^p$. Let the binary vector $\mathbf{c}(\mathbf{x}_j^p) \in \{0, 1\}^N$ represent the selection vector, such that $(\mathbf{x}_i, \hat{y}_i)$ is selected for $\mathbf{x}_j^p$ when $c_i(\mathbf{x}_j^p) = 1$.

Now, we convert the intractable optimization problem in Eq. (1) with the following approximations:

**(i)** The sample mean is used as an approximation of the first term of the objective function:

$$\mathbb{E}_{\mathbf{x}^p \sim P_X} \left[ \mathcal{L}(f^*(\mathbf{x}^p), g^*_{\theta(\mathbf{x}^p)}(\mathbf{x}^p)) \right] \simeq \frac{1}{M} \sum_{j=1}^{M} \mathcal{L}(f^*(\mathbf{x}_j^p), g^*_{\theta(\mathbf{x}_j^p)}(\mathbf{x}_j^p))$$

**(ii)** The second term of the objective function, which represents the average selection probability, is approximated as the average number of selected instances:

$$\mathbb{E}_{\mathbf{x}^p, \mathbf{x} \sim P_X} \left[ h_\phi(\mathbf{x}^p, \mathbf{x}, f^*(\mathbf{x})) \right] \simeq \frac{1}{MN} \sum_{j=1}^{M} \sum_{i=1}^{N} |c_i(\mathbf{x}_j^p)|$$

**(iii)** The objective of the constraint term is approximated using the sample mean of the training loss as

$$\mathbb{E}_{\mathbf{x} \sim P_X} \left[ h_\phi(\mathbf{x}^p, \mathbf{x}, f^*(\mathbf{x})) \cdot \mathcal{L}_g(f^*(\mathbf{x}), g_\theta(\mathbf{x})) \right] \simeq \frac{1}{N} \sum_{i=1}^{N} \left[ c_i(\mathbf{x}_j^p) \cdot \mathcal{L}_g(f^*(\mathbf{x}_i), g_\theta(\mathbf{x}_i)) \right]$$

The converted tractable optimization problem becomes:

$$\min_{h_\phi} \quad \frac{1}{M} \sum_{j=1}^{M} \left[ \mathcal{L}(f^*(\mathbf{x}_j^p), g^*_{\theta(\mathbf{x}_j^p)}(\mathbf{x}_j^p)) + \lambda \frac{1}{N} \sum_{i=1}^{N} |c_i(\mathbf{x}_j^p)| \right]$$
$$\text{s.t.} \quad g^*_{\theta(\mathbf{x}_j^p)} = \arg\min_{g_\theta} \frac{1}{N} \sum_{i=1}^{N} \left[ c_i(\mathbf{x}_j^p) \cdot \mathcal{L}_g(f^*(\mathbf{x}_i), g_\theta(\mathbf{x}_i)) \right] \text{ where } c_i(\mathbf{x}_j^p) \sim Ber(h_\phi(\mathbf{x}_j^p, \mathbf{x}_i, f^*(\mathbf{x}_i))).$$
$$(2)$$

The sampler block yields a non-differential objective as the optimization is over $\mathbf{c}(\mathbf{x}_j^p) \in \{0, 1\}^N$ - weighted instances, and we cannot use conventional gradient descent-based optimization to solve the above optimization problem. Motivated by its successful applications (Ranzato et al., 2015; Zaremba & Sutskever, 2015; Zhang & Lapata, 2017), we adapt the policy-gradient based REINFORCE algorithm (Williams, 1992) such that the selection action[1] is rewarded by its impact on performance. We consider the loss function

$$l(\phi) = \frac{1}{M} \sum_{j=1}^{M} \left[ \mathcal{L}(f^*(\mathbf{x}_j^p), g^*_{\theta(\mathbf{x}_j^p)}(\mathbf{x}_j^p)) + \lambda \frac{1}{N} \sum_{i=1}^{N} |c_i(\mathbf{x}_j^p)| \right]$$

---

[1]States are the features of input instances, actions are the selection vectors from $h_\phi$ (policy) that selects the most valuable samples, and reward is the fidelity of the locally interpretable model compared to the black box model which depends on the input features (state) and the selection vector (action).

as the reward given the state and action for the selection policy[2].

Correspondingly, $\rho_\phi(\mathbf{x}_j^p, \mathbf{c}(\mathbf{x}_j^p))$ is the probability mass function for $\mathbf{c}(\mathbf{x}_j^p)$ given $h_\phi(\cdot)$:

$$\rho_\phi(\mathbf{x}_j^p, \mathbf{c}(\mathbf{x}_j^p)) = \prod_{i=1}^{N} \left[ h_\phi(\mathbf{x}_j^p, \mathbf{x}_i, f^*(\mathbf{x}_i))^{c_i(\mathbf{x}_j^p)} \cdot (1 - h_\phi(\mathbf{x}_j^p, \mathbf{x}_i, f^*(\mathbf{x}_i)))^{1-c_i(\mathbf{x}_j^p)} \right].$$

To apply the REINFORCE algorithm, we directly compute its gradient with respect to $\phi$:

$$\nabla_\phi \hat{l}(\phi) = \frac{1}{M} \sum_{j=1}^{M} \left[ \mathcal{L}(f^*(\mathbf{x}_j^p), g^*_{\theta(\mathbf{x}_j^p)}(\mathbf{x}_j^p)) + \lambda \frac{1}{N} \sum_{i=1}^{N} |c_i(\mathbf{x}_j^p)| \right] \nabla_\phi \log \rho_\phi(\mathbf{x}_j^p, \mathbf{c}(\mathbf{x}_j^p)).$$

Bringing all this together, we update the parameters of the instance-wise weight estimator $\phi$ with the following steps (bi-level optimization) repeated until convergence:

**(i)** Estimate instance-wise weights $w_i(\mathbf{x}_j^p) = h_\phi(\mathbf{x}_j^p, \mathbf{x}_i, \hat{y}_i)$ and instance selection vector $c_i(\mathbf{x}_j^p) \sim \mathrm{Ber}(w_i(\mathbf{x}_j^p))$ for each training and probe instance in a mini-batch ($N_{mb}$ is the number of samples in a mini batch).
**(ii)** Optimize the locally interpretable model with the selection for each probe instance:

$$g^*_{\theta(\mathbf{x}_j^p)} = \arg\min_{g_\theta} \sum_{i=1}^{N_{mb}} \left[ c_i(\mathbf{x}_j^p) \cdot \mathcal{L}_g(f^*(\mathbf{x}_i), g_\theta(\mathbf{x}_i)) \right] \tag{3}$$

**(iii)** Update the instance-wise weight estimation model (where $\alpha > 0$ is a learning rate):

$$\phi \leftarrow \phi - \frac{\alpha}{M} \sum_{j=1}^{M} \left[ \mathcal{L}(f^*(\mathbf{x}_j^p), g^*_{\theta(\mathbf{x}_j^p)}(\mathbf{x}_j^p)) + \lambda \frac{1}{N} \sum_{i=1}^{N} |c_i(\mathbf{x}_j^p)| \right] \cdot \nabla_\phi \log \rho_\phi(\mathbf{x}_j^p, \mathbf{c}(\mathbf{x}_j^p)) \tag{4}$$

Pseudo-code of the LIMIS training is in Algorithm. 1. We stop training LIMIS algorithm if there are no fidelity improvements. Hyper-parameters are optimized to maximize the validation fidelity.

---

**Algorithm 1** LIMIS Training

---

**Input**: Training data ($\mathcal{D}$), probe data ($\mathcal{D}^p$), black-box model ($f^*$)

1: **Initialize** $h_\phi$.
2: **Construct auxiliary data** ($\hat{\mathcal{D}}, \hat{\mathcal{D}}^p$): $\hat{\mathcal{D}} = \{(\mathbf{x}_i, f^*(\mathbf{x}_i))\}_{i=1}^{N}$, $\hat{\mathcal{D}}^p = \{(\mathbf{x}_j^p, f^*(\mathbf{x}_j^p))\}_{j=1}^{M}$
3: **while** $h_\phi$ is not converged **do**
4:      Estimate $w_i(\mathbf{x}_j^p) = h_\phi(\mathbf{x}_j^p, \mathbf{x}_i, \hat{y}_i)$
5:      Sample $c_i(\mathbf{x}_j^p) \sim \mathrm{Ber}(w_i(\mathbf{x}_j^p))$
6:      Optimize locally interpretable models with $c_i(\mathbf{x}_j^p)$ using Eq. (3)
7:      Update $h_\phi$ using Eq. (4)
8: **end while**

**Output**: Trained instance-wise weight estimator ($h_\phi^*$)

---

● **Stage 3 − Interpretable inference**: Unlike training, we use a *fixed* instance-wise weight estimator without the sampler. Note that we use the probabilistic selection for encouraging the exploration during the training. At inference time, exploration is no longer needed; it would be better to minimize the randomness to maximize the fidelity of the locally interpretable models. Given the test instance $\mathbf{x}^t$, we obtain the selection probabilities from the instance-wise weight estimator, and using these as the weights, we fit the locally interpretable model via weighted optimization. The outputs of the fitted model are the instance-wise predictions and the corresponding explanations (e.g. coefficients for a linear model). Pseudo-code of the LIMIS inference is in Algorithm. 2.

---

[2]Other desired properties, such as robustness of explanations against input perturbations, can be further added to the reward – the flexibility constitutes one of the major advantages.

---

**Algorithm 2** LIMIS Inference

---

**Input**: Training data ($\mathcal{D}$), test sample ($\mathbf{x}^t$), trained instance-wise weight estimator ($h_\phi^*$)

1: Estimate $w_i(\mathbf{x}^t) = h_\phi^*(\mathbf{x}_t, \mathbf{x}_i, \hat{y}_i)$
2: Optimize locally interpretable model using instance-wise weights $w_i(\mathbf{x}^t)$ via weighted optimization:
3: $g_{\theta(\mathbf{x}^t)}^* = \arg\min_{g_\theta} \sum_{i=1}^N w_i(\mathbf{x}^t) \cdot \mathcal{L}_g(f^*(\mathbf{x}_i), g_\theta(\mathbf{x}_i))$

**Output**: Predictions ($g_{\theta(\mathbf{x}^t)}^*(\mathbf{x}^t)$), explanations ($g_{\theta(\mathbf{x}^t)}^*$), and instance-wise weights $\{w_i(\mathbf{x}^t)\}_{i=1}^N$

---

### 3.2 Computational cost

As a representative and commonly used example, consider a linear ridge regression (RR) model as the locally interpretable model, which has a computational complexity of $\mathcal{O}(d^2 N) + \mathcal{O}(d^3)$ to fit, where $d$ is the number of features and $N$ is the number of training instances. When $N \gg d$ (which is often the case in practice), the training computational complexity is approximated as $\mathcal{O}(d^2 N)$ (Tan, 2018).

**Training:** Given a pre-trained black-box model, Stage 1 involves running inference $N$ times and the total complexity is determined by the black-box model. Unless the black-box model is very complex, the computational cost of Stage 1 is much smaller than Stage 2. At Stage 2, we iteratively train the instance-wise weight estimator and fit the locally interpretable model using weighted optimization. Therefore, the computational complexity is $\mathcal{O}(d^2 N N_I)$ where $N_I$ is the number of iterations (typically $N_I < 10,000$ until convergence). Thus, the training complexity scales roughly linearly with the number of training instances.

**Interpretable inference:** To infer with the locally interpretable model, we need to fit the locally interpretable model after obtaining the instance-wise weights from the trained instance-wise weight estimator. For each testing instance, the computational complexity is $\mathcal{O}(d^2 N)$.

Experimental results on the computational cost for both training and inference can be found in Sect 5.3.

## 4 Synthetic Data Experiments

Evaluations of explanation quality are challenging on real-world datasets due to the absence of ground-truth explanations. Therefore, we initially perform experiments on synthetic datasets with known ground-truth explanations to directly evaluate how well the locally interpretable models can recover the underlying reasoning behind outputs.

We construct three synthetic datasets that have different local behaviors in different input regimes. The 11-dimensional input features $\mathbf{X}$ are sampled from $\mathcal{N}(0, I)$ and $Y$ are determined as follows:

• Syn1: $Y = X_1 + 2X_2$ if $X_{10} < 0$ & $Y = X_3 + 2X_4$ if $X_{10} \geq 0$,

• Syn2: $Y = X_1 + 2X_2$ if $X_{10} + e^{X_{11}} < 1$ & $Y = X_3 + 2X_4$ if $X_{10} + e^{X_{11}} \geq 1$,

• Syn3: $Y = X_1 + 2X_2$ if $X_{10} + (X_{11})^3 < 0$ & $Y = X_3 + 2X_4$ if $X_{10} + (X_{11})^3 \geq 0$.

$Y$ is directly dependent to $X_1, ..., X_4$ and not directly dependent to $X_{10}, X_{11}$. However, $X_{10}, X_{11}$ determine how $Y$ are dependent on $X_1, ..., X_4$. For instance, in Syn1 dataset, $Y$ is directly dependent with $X_1, X_2$ if $X_{10}$ is negative. If $X_{10}$ is positive, $Y$ is directly dependent with $X_3, X_4$ but independent of $X_1, X_2$. Additional results with non-linear feature-label relationships can be found in Appendix. H.7.

We directly use the ground truth function as the black-box model instead of a fitted nonlinear black-box model to solely focus on LIMIS performance, decoupling from the nonlinear black-box model fitting performance. We quantify how well locally interpretable modeling can capture the underlying local function behavior using the Absolute Weight Difference (AWD) metric: AWD $= ||\mathbf{w} - \hat{\mathbf{w}}||$, where $\mathbf{w}$ is the ground truth linear coefficients to generate $Y$ given $X$ and $\hat{\mathbf{w}}$ is the estimated coefficient from the linear locally interpretable model (RR in our experiments). To make the experiments consistent and robust, we use the *probe sample* as the criteria to determine the ground-truth local dynamics ($\mathbf{w}$). We report the results over 10 independent

runs with 2,000 samples per each synthetic dataset. Additional results can be found in the Appendix. E, F, and G.

## 4.1 Recovering local function behavior

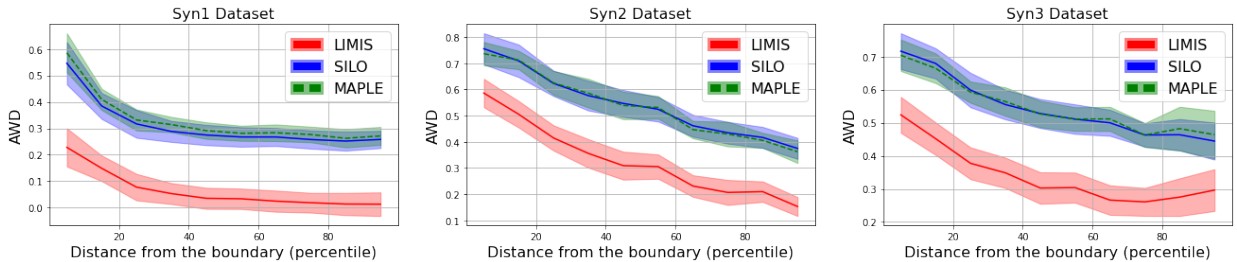

Figure 3: Mean AWD (aggregated per uniformly divided x-axis bin) with 95% confidence intervals (of 10 independent runs) on three synthetic datasets (y-axis) vs. the percentile distance from the boundary where the local function behavior change (x-axis), e.g. $X_{10} = 0$ for Syn1. We exclude LIME due to its poor performance (its AWD is higher than 1.6 in all cases distance regimes for all datasets). LIME results and the scatter plots of LIMIS can be found in the appendix.

We compare LIMIS to LIME (Ribeiro et al., 2016), SILO (Bloniarz et al., 2016), and MAPLE (Plumb et al., 2018). Fig. 3 shows that LIMIS significantly outperforms other methods in discovering the local function behavior on all three datasets, in different regimes. Even the decision boundaries are non-linear (Syn2 and Syn3), LIMIS can efficiently learn them, beyond the capabilities of the linear RR model. LIME fails to recover the local function behavior as it uses the Euclidean distance and cannot distinguish the special properties of the features. SILO and MAPLE only use the relevant variables for the predictions; thus, it is difficult for them to discover the decision boundary that depends on other variables, independent of the predictions.

## 4.2 The impact of the number of selected instances

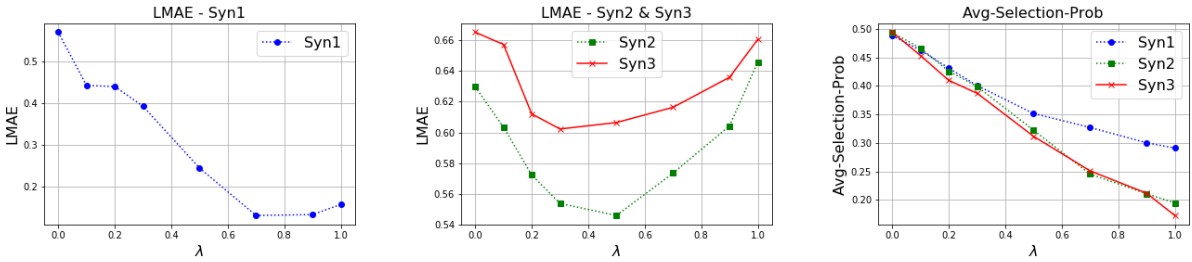

Figure 4: Fidelity (in LMAE) and average selection probability of training samples (y-axis) vs. $\lambda$ (x-axis).

Optimal distillation in LIMIS is enabled by using a small subset of training instances to fit the low-capacity locally interpretable model. The number of selected instances is controlled by $\lambda$ – if $\lambda$ is high/low, LIMIS penalizes more/less, thus less/more instances are selected to fit the locally interpretable model. We analyze the efficacy of $\lambda$ in controlling the likelihood of selection and the fidelity. Fig. 4 (left and middle) demonstrates the clear relationship between $\lambda$ and the fidelity. If $\lambda$ is too large, LIMIS selects insufficient number of instances; thus, the fitted locally interpretable model is less accurate (due to underfitting). If $\lambda$ is too small, LIMIS is not sufficiently encouraged to select the most relevant instances related to the local dynamics to fit the ridge regression model, thus is more prone to learning feature relationships that may generalize poorly. To achieve the optimal $\lambda$, we conduct cross-validation and select $\lambda$ with the best validation fidelity. Fig. 4 (right) shows the average selection probability of the training instances for each $\lambda$. As $\lambda$ increases, the average selection probabilities decrease due to the higher penalty on the number of selected instances. Even using a

small portion of training instances, LIMIS can accurately distill the predictions into locally interpretable models, which is crucial to understand the predictions using the most relevant instances.

### 4.3 Comparison to differentiable baselines

Table 1: AWD comparisons on three synthetic datasets with different number of train samples ($N$).

| Number of train samples | $N = 500$ | | | | $N = 1000$ | | | | $N = 2000$ | | | |
|---|---|---|---|---|---|---|---|---|---|---|---|---|
| Datasets | Syn1 | Syn2 | Syn3 | **Avg.** | Syn1 | Syn2 | Syn3 | **Avg.** | Syn1 | Syn2 | Syn3 | **Avg.** |
| LIMIS | .5531 | .5869 | .6512 | .5971 | **.2129** | **.4289** | .5527 | **.3982** | **.1562** | **.3325** | **.3920** | **.2936** |
| Gumbel-softmax | **.4177** | .5017 | **.5953** | **.5049** | .2712 | .4511 | .5405 | .4209 | .1698 | .3655 | .4217 | .3190 |
| STE | .4281 | **.4941** | .6001 | .5074 | .2688 | .4407 | **.5372** | .4156 | .1717 | .3601 | .4307 | .3208 |
| L2R | .6758 | .6607 | .6903 | .6756 | .6989 | .6412 | .6217 | .6539 | .7532 | .7283 | .7506 | .7440 |

We compare LIMIS to three baselines that have differentiable objectives for data weighting in Table 1: (1) Gumbel-softmax (Jang et al., 2016)[3], (2) straight-through estimator (STE) (Bengio et al., 2013), (3) Learning to Reweight (L2R) (Ren et al., 2018).

LIMIS utilizes the sampling procedure; thus, the objective (loss) function is non-differentiable. In that case, we cannot train the model in an end-to-end way using the stochastic gradient descent (SGD). Instead, there are multiple ways to train with non-differentiable objectives. In the LIMIS framework, we use the REINFORCE algorithm to train the model with non-differentiable objective. Gumbel-softmax is another approximation method to convert non-differentiable sampling procedure to the differentiable softmax outputs. STE is replacing the sampling procedure with direct weighted optimization. Both STE and Gumbel-softmax baselines are differentiable approaches and we can optimize the models in an end-to-end way via SGD.

We observe that Gumbel-softmax and STE converge faster but to a suboptimal solution, due to under-exploration. L2R overfits to the fidelity metric and cannot guide weighting of the instances accurately, yielding poor AWD. Because L2R learns the same weights across all instances, whereas LIMIS uses an instance-wise weight estimator to learn instance-wise weights separately for each probe instance. In Table 1, Gumbel-softmax and STE models outperform LIMIS only if $N = 500$ (in the regime of extremely small number of training instances), given their favorable inductive bias with gradient-descent based optimization (that also yields faster convergence). However, with $N = 1000, 2000$, they underperform LIMIS due to the under-exploration. More specifically, the average performance improvements with LIMIS is 4.2%, 6.4% (with respect to $N = 1000, 2000$) in comparison with the best alternative. As seen in Fig. 9, the performance gap between LIMIS and alternatives increases as the number of training samples increases.

## 5 Real-world Data Experiments

We next study LIMIS on 3 real-world regression datasets: (1) Blog Feedback, (2) Facebook Comment, (3) News Popularity; and 2 real-world classification datasets: (4) Adult Income, (5) Weather. We use raw data after normalizing each feature to be in $[0, 1]$, using standard Min-Max scaler and apply one-hot encoding to categorical features. We focus on black-box models that are shown to yield strong performance on target tasks. We implement the instance-wise weight estimator as an MLP with tanh activation. Its hyperparameters are optimized using cross-validation (5-layer MLP with 100 hidden units performs reasonably-well across all datasets). Model details on the data and hyperparameters can be found in the Appendix. D and A.

### 5.1 Performance comparisons

We evaluate the performance on disjoint testing sets $\mathcal{D}^t = \{(\mathbf{x}_k^t, y_k^t)\}_{k=1}^{L} \sim \mathcal{P}$ and report the results over 10 independent runs. For fidelity, we compare the outputs (predicted values for regression and logits for classification) of the locally interpretable models and the black-box model, using Nash-Sutcliffe Efficiency (NSE) (Nash & Sutcliffe, 1970) For the prediction performance, we use Mean Absolute Error (MAE) for

---

[3]We set Gumbel-softmax temperature as 0.5; we do not use temperature annealing due to the training instability.

regression and Average Precision Recall (APR) for classification. Details on the metrics can be found in Appendix. B.

Table 2: Fidelity (metric: NSE, higher is better) and prediction performance (metric: MAE, lower is better / APR, higher the better) on regression/classification datasets, using RR/DT as the locally interpretable model while explaining the black box models: XGBoost (Chen & Guestrin, 2016), LightGBM (Ke et al., 2017), Random Forests (RF) (Breiman, 2001) and Multi-layer Perceptron (MLP). 'Original' represents the performance of the original black-box model that the locally-interpretable modeling is applied on. We also show the performance of RR/ DT (in terms of MAE/APR) as a globally-interpretable model under the data name. Red: performance worse than globally-interpretable RR/DT and the negative NSE. **Bold**: best results.

| **Regression Datasets** | Models | XGBoost | | LightGBM | | MLP | | RF | |
|---|---|---|---|---|---|---|---|---|---|
| (Ridge Regression) | Metrics | MAE | NSE | MAE | NSE | MAE | NSE | MAE | NSE |
| **Blog**
(8.420) | Original | 5.131 | 1.0 | 4.965 | 1.0 | 4.893 | 1.0 | 5.203 | 1.0 |
| | LIMIS | **5.289** | **.8679** | **4.971** | **.9069** | **4.994** | **.7177** | **4.993** | **.8573** |
| | LIME | 9.421 | .3440 | 10.243 | .3019 | 10.936 | -.2723 | 19.222 | -.2143 |
| | SILO | 6.261 | .0005 | 6.040 | .2839 | 5.413 | .4274 | 6.610 | .4500 |
| | MAPLE | 5.307 | .8248 | 4.981 | .8972 | 5.012 | .5624 | 5.058 | .8471 |
| **Facebook**
(24.64) | Original | 24.18 | 1.0 | 20.22 | 1.0 | 18.36 | 1.0 | 30.09 | 1.0 |
| | LIMIS | **22.92** | **.7071** | 24.84 | **.4268** | **20.23** | **.5495** | **22.65** | **.4360** |
| | LIME | 35.20 | .2205 | 38.19 | .2159 | 38.82 | .2463 | 51.77 | .1797 |
| | SILO | 31.41 | -.4305 | 39.10 | -1.994 | 22.35 | .3307 | 42.05 | -.7929 |
| | MAPLE | 23.28 | .6803 | 41.86 | -3.233 | 24.77 | -.1721 | 44.75 | -1.078 |
| **News**
(.2989) | Original | 2995 | 1.0 | 3140 | 1.0 | 2255 | 1.0 | 3378 | 1.0 |
| | LIMIS | **2958** | **.7534** | **2957** | **.5936** | 2260 | **.9761** | **2396** | **.6523** |
| | LIME | 5141 | -.2467 | 6301 | -2.008 | 2289 | .5030 | 9435 | -7.477 |
| | SILO | 3069 | .4547 | 3006 | .4025 | **2257** | .9617 | 3251 | .3816 |
| | MAPLE | 2967 | .7010 | 3005 | .3963 | 2259 | .9534 | 3060 | .5901 |
| **Classification Datasets** | Models | XGBoost | | LightGBM | | MLP | | RF | |
| (Decision Tree) | Metrics | APR | NSE | APR | NSE | APR | NSE | APR | NSE |
| **Adult**
(.6388) | Original | .8096 | 1.0 | .8254 | 1.0 | .7678 | 1.0 | .7621 | 1.0 |
| | LIMIS | **.8011** | **.9889** | **.8114** | **.9602** | .7710 | .9451 | **.7881** | **.8788** |
| | LIME | .6211 | .5009 | .6031 | .3798 | .4270 | .2511 | .6166 | .3833 |
| | SILO | .8001 | .9869 | .8107 | .9583 | .7708 | **.9470** | .7833 | .8548 |
| | MAPLE | .7928 | .9794 | .8034 | .9405 | **.7719** | .9410 | .7861 | .8622 |
| **Weather**
(.5838) | Original | .7133 | 1.0 | .7299 | 1.0 | .7205 | 1.0 | .7274 | 1.0 |
| | LIMIS | **.7071** | **.9734** | **.7118** | **.9601** | **.7099** | **.9124** | **.7102** | **.9008** |
| | LIME | .6179 | .7783 | .6159 | .6913 | .5651 | .3417 | .6209 | .3534 |
| | SILO | .6991 | .9680 | .7052 | .9452 | .6997 | .8864 | .7042 | .8398 |
| | MAPLE | .6973 | .9675 | .7056 | .9446 | .6983 | .8856 | .6983 | .8856 |

Table 2 shows that for regression tasks, the performance of globally interpretable RR (trained on the entire dataset from scratch) is much worse than complex black-box models, underlining the importance of non-linear modeling. Locally interpretable modeling with LIME, SILO and MAPLE yield significant performance degradation compared to the original black-box model. In some cases (e.g. on Facebook), the performance of previous work is even worse than the globally interpretable RR, undermining the use of locally interpretable modeling. In contrast, LIMIS achieves consistently high prediction performance and significantly outperforms RR. Table 2 also compares the fidelity in terms of NSE. We observe that NSE is negative for some cases (e.g. LIME on Facebook data), implying that output of the locally interpretable model is even worse than the constant mean value estimator. On the other hand, LIMIS achieves high NSE consistently across all datasets with all black-box models. Table 2 also shows the performance on classification tasks using shallow regression

DTs as the locally interpretable model (Regression DTs model outputs logits for classification.). Among the locally interpretable models, LIMIS often achieves the best APR and NSE, underlining its strength in distilling the predictions of the black-box model accurately. In some cases, the benchmarks (especially LIME) yield worse prediction performance than the globally interpretable model, DT. Additional results can be found in the Appendix G.

## 5.2 Local generalization of explanations

For locally interpretability modeling, local generalization of explanations is very important, as one expect a similar behavior around a meaningful vicinity of a sample. To quantify the local generalization of explanations, we include evaluations with neighborhood metrics (Plumb et al., 2019), which give insights on the explanation quality at nearby points. We show the results on two regression datasets (Blog and Facebook) with two black-box models (XGBoost and LightGBM) and evaluate them in terms of neighborhood LMAE (Local MAE) and pointwise LMAE. Here, the difference would be a measure of local generalization, i.e. how reliable the explanations are against input changes. Neighborhood LMAE is defined as $\mathbb{E}_{\mathbf{x}^t \sim \mathcal{P}_X, n \sim \mathcal{N}(0, \sigma I)} ||g^*_{\theta(\mathbf{x}^t)}(\mathbf{x}^t + n) - f^*(\mathbf{x}^t + n)||_1$ where we set $\sigma = 0.1$ as the neighborhood vicinity (with standard normalization for the inputs). Pointwise LMAE is defined as $\mathbb{E}_{\mathbf{x}^t \sim \mathcal{P}_X} ||g^*_{\theta(\mathbf{x}^t)}(\mathbf{x}^t) - f^*(\mathbf{x}^t)||_1$. Details on the metrics can be found in Appendix. B. Additional results can be also found in Appendix. H.4.

Table 3: Prediction performance (metric: *neighborhood* LMAE and *pointwise* LMAE, lower is better) on regression datasets, using RR as the locally interpretable model while explaining the black box models: XGBoost (Chen & Guestrin, 2016) and LightGBM (Ke et al., 2017). **Bold**: best results.

| Datasets | Models | XGBoost | | | LightGBM | | |
|---|---|---|---|---|---|---|---|
| (RR) | Metrics | *Neighbor* LMAE | *Pointwise* LMAE | Diff | *Neighbor* LMAE | *Pointwise* LMAE | Diff |
| **Blog** | LIMIS | **.8894** | **.8679** | 2.48% | **1.217** | **1.135** | 7.22% |
| | LIME | 6.872 | 6.534 | 5.17% | 8.233 | 8.037 | 2.44% |
| | SILO | 2.368 | 2.220 | 6.68% | 3.119 | 3.046 | 2.40% |
| | MAPLE | 1.007 | .9690 | 3.96% | 1.442 | 1.416 | 1.87% |
| **Facebook** | LIMIS | **6.533** | **6.394** | 2.18% | **8.250** | **8.217** | 0.41% |
| | LIME | 32.82 | 32.57 | 0.77% | 34.85 | 33.70 | 3.31% |
| | SILO | 19.82 | 19.51 | 1.60% | 30.60 | 30.07 | 1.79% |
| | MAPLE | 8.189 | 7.664 | 6.86% | 31.32 | 31.25 | 0.25% |

As can be seen in Table 3, LIMIS's superior performance is still apparent in neighborhood MAE. For instance, LIMIS achieves 25.56 in neighborhood metric (with MAE) which is better than the results with MAPLE (42.21) and LIME (39.51) using Facebook data and LightGBM model (over 10 independent runs). Note that the differences between pointwise and neighborhood fidelity metrics with LIMIS are negligible across other datasets and black-box models. This shows that the performance of LIMIS is locally generalizable and reliable, which is the main objective of locally interpretable modeling.

## 5.3 Computational time

We quantify the computational time on the largest experimented dataset, Facebook Comments, that consists $\sim 600{,}000$ samples. On a single NVIDIA V100 GPU (without any hardware optimizations), LIMIS yields a training time of less than 5 hours (including Stage 1, 2 and 3) and an interpretable inference time of less than 10 seconds per testing instance. On the other hand, LIME results in much longer interpretable inference time, around 30 seconds per a testing instance, due to acquiring a large number of black-box model predictions for the input perturbations, while SILO and MAPLE show similar computational time with LIMIS.

## 6 LIMIS Explainability Use Cases

In this section, we showcase explainability use-cases of LIMIS for human-in-the-loop AI deployments. LIMIS can distill complex black-box models into explainable surrogate models, such as shallow DTs or linear

regression. These surrogate models are explainable and can bring significant value to many applications where exact and concise input-output mapping would be needed. As the fidelity of LIMIS is very high, the users would have high trust in the surrogate models. Still, the outputs of the LIMIS should not be treated as the ground-truth interpretations (local dynamics).

## 6.1 Sample-wise feature importance

Discovering the importance of features for a prediction is one of the most commonly-used explanation tasks. Locally interpretable modeling can enable this capability by using a linear model as the surrogate, since the coefficients of linear models can directly tell how features are combined for a prediction.

To highlight this capability, we consider LIMIS with RR on UCI Adult Income, shown in Fig. 5. Here, we use XGBoost as the black-box model along with the locally interpretable RR. We focus on visualizing the feature importance (the absolute weights of the fitted locally interpretable RR model) for various important subsets, to analyze common decision making drivers for them. Specifically, we focus on 5 subgroups, divided based on (a) Age, (b) Gender, (c) Marital status, (d) Race, (e) Education.

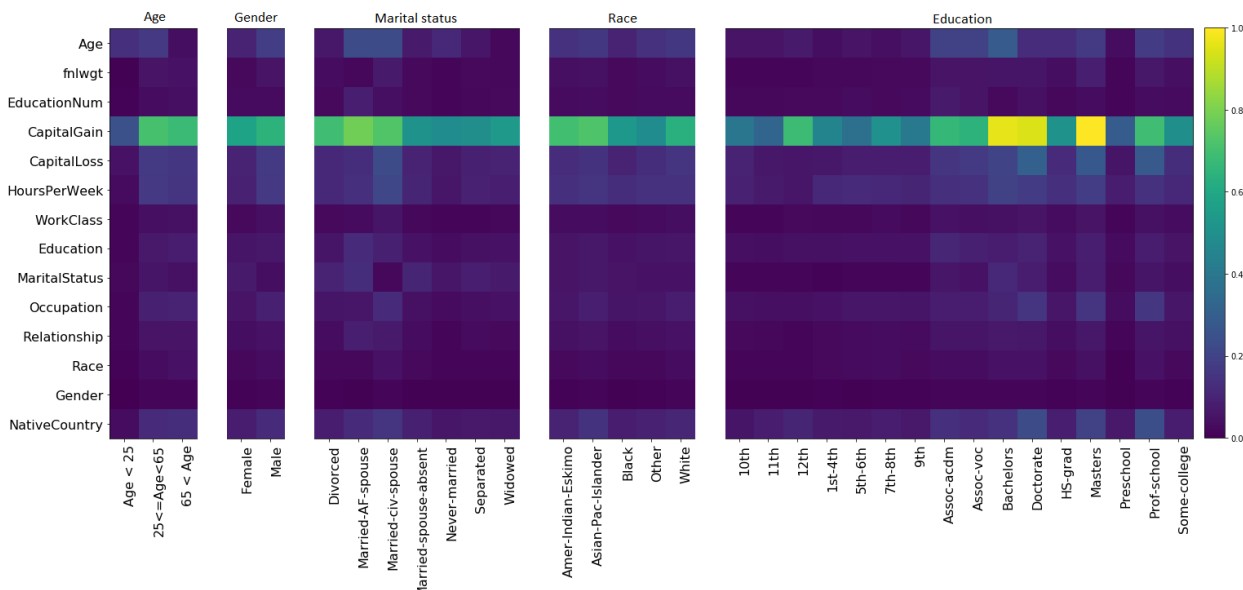

Figure 5: Feature importance (denoted with the colors) discovered by LIMIS as the absolute weights of the fitted locally interpretable RR model, on UCI Adult Income, for 5 types of subgroups: (a) Age, (b) Gender, (c) Marital status, (d) Race, (e) Education.

We observe that for age-based subgroups, more importance is attributed to capital gain for older people (age $> 25$) compared to young people (age $\leq 25$), as intuitively expected given that older people often have more savings (Wolff, 2015). For education-based subgroups, capital gain/loss, occupation, and native countries are attributed to be more important for highly-educated people (Doctorate, Prof-school, and Masters graduates), as also intuitively expected as the income gaps tend to get wider with education levels (Nordin et al., 2010; Sullivan & Wolla, 2017). LIMIS can also be used for dependency assessments for black-box models by analyzing the importance of sensitive attribute features locally to see whether any would play a significant role in decision making. For this example, some attributes (such as gender, marital status and race) are not observed to have high importance for most subgroups.

These exemplify how local explanations from LIMIS can bring value to users, particularly with a linear interpretable model where the learned weights can readily provide insights on how features affect the predictions. We provide more results in the Appendix. G.7.

## 6.2 Suggesting actionable input modifications to alter decisions

In many applications, it is desired to understand what it would take to alter the decision of a model. For example, after rejection of a loan application, the applications would want to understand what they can change to get it accepted, or after a disease diagnosis, doctors and patients would want to understand the suggestions on what can be changed about the patients health to reverse the diagnosis prediction.

The fundamental benefit of locally interpretable modeling is that it allows understanding the how features are affecting the prediction for an instance, via an interpretable model that yields the precise relationship. High fidelity of LIMIS with simple interpretable models enable this capability effectively.

| No | Key characteristics | Prediction | Suggestion for $>\$50K$ income |
|---|---|---|---|
| 1 | Education: High-school, No capital gain | $<\$50K$ | Get Masters & increase capital gains to 6K |
| 2 | No capital gain, Hours per week: 40 | $>\$50K$ | - |
| 3 | Age: 33, Education year: 13, Married | $>\$50K$ | - |
| 4 | Age: 44, Job: Craft-repair | $<\$50K$ | Increase capital gain by $6K$ |
| 5 | Job: Local-gov, Education: HS, Hours per week: 40 | $<\$50K$ | Change job to Federal-gov |
| 6 | Capital loss: 23K, Job: Sales, Education: College | $<\$50K$ | Decrease the capital loss to 9K |
| 7 | Hours per week: 26, Job: Sales | $<\$50K$ | Change job to Tech support |
| 8 | Capital gain: 15K, Masters, Age: 51 | $>\$50K$ | Increase the capital gain to 10K |
| 9 | Capital loss: 17K, Hours per week: 40 | $<\$50K$ | Reduce the capital loss to 11K |
| 10 | Age: 38, Occupation: Exec managerial | $<\$50K$ | - |

Table 4: For ten individuals explanations given by LIMIS using shallow DT on UCI Adult dataset are shown. The individual characteristics are based on the DT and the suggestions are obtained with the goal of making the locally interpretable model prediction as $>\$50K$, by inspecting the fitted DT.

For this demonstration, shown in Table. 4, we consider LIMIS with shallow DTs (with a depth of 3) as interpretable models, on the UCI Adult Income dataset. LIMIS is first trained on the entire training data, and then, for some test instance, LIMIS is used to extract local explanations on the predictions of black-box model, XGBoost. In addition, we consider the question of 'what's the suggested minimum change to alter the decision?' and utilize the fitted shallow DT for this purpose. Essentially, our approach to find the modification suggestion relies on traversing the DT upwards from the leaf, and finding the nearest (in terms of being closest in edge distance) node condition that would yield the opposite prediction. We specifically focus on suggestions to increase the income prediction from low ($<\$50K$) to high income ($>\$50K$)

In most cases, we observe that the suggestions are consistent with the expectations (Nordin et al., 2010; Sullivan & Wolla, 2017; Wolff, 2015). For example, better investment outcomes, higher paying jobs and additional education are among common suggestions. When the inputs are modified with these changes, the black-box model predictions change from $<\$50K$ to $>\$50K$, for all cases exemplified here, underlining the accuracy of the suggestions.

## 7 Conclusions

We propose a novel method for locally interpretable modeling of pre-trained black-box models, called LIMIS. LIMIS selects a small number of valuable instances and uses them to train a low-capacity locally interpretable model. The selection mechanism is guided with a reward obtained from the similarity of predictions of the locally interpretable model and the black-box model, defined as fidelity. LIMIS near-matches the performance of black-box models, and significantly outperforms alternatives, consistently across various datasets and for various black-box models. We demonstrate the high-fidelity explanations provided by LIMIS can be highly useful to gain insights about the task and to understand what would modify the model's outcome.

## 8 Broader Impact

Interpretability is critical, to increase the reach of AI to many more use cases compared to its reach today, in a reliable way, by showing rationale behind decisions, eliminating biases, improving fairness, enabling detection of systematic failure cases, and providing actionable feedback to improve models (Rudin, 2018). We introduce a novel method, LIMIS, that tackles interpretability via instance-wise weighted training to provide local explanations. LIMIS provides highly faithful, and easy-to-digest explanations to humans. Applications of LIMIS can span understanding instance-wise local dynamics, building trust by explaining the constituent components behind the decisions and enabling actionable insights such as manipulating outcomes. For scenarios such as medical treatment planning, where the input variables can be controlled based on the feedback from the output responses, interpretable local dynamics can be highly valuable for manipulating outcomes (Saria et al., 2018).

As one limitation, the proposed instance-wise weight estimator is indeed a black-box model and difficult to interpret without post-hoc interpretable methods. However, the main value proposition of LIMIS is that the final output from the locally interpretable model, is fully interpretable and the users can utilize the final output for understanding the rationale behind the local decision making process of the black-box model. With our experiments, we show that the fidelity metrics of the locally interpretable models of LIMIS are high, in other words, for each sample, they approximate the black-box model functionality very well.

Broadly, there are many different forms of explainable AI approaches, from single interpretable models to post-hoc methods for complex black-box models. LIMIS is not an alternative to all of them, but it specifically provides the locally interpretable modeling capability, which has numerous impactful use cases in real-world AI deployments, including Finance, Healthcare, Policy, Law, Recruiting, Physical Sciences, and Retail.

We demonstrate that LIMIS provides the local interpretability capabilities similar to other notable methods like LIME, while achieving much higher fidelity, as a strong quantitative evidence for its utility. Still, the outputs of the LIMIS framework should not be treated as the ground truth interpretation of the black-box models. Large-scale human subject evaluations can further add confidence in the capabilities of LIMIS and the quality of its explanations – we leave this important aspect to future work.

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

# A    Hyper-parameters of the predictive models

In this paper, we use 8 different predictive models. For each predictive model, the corresponding hyper-parameters used in the experiments are as follows:

- **XGBoost (Chen & Guestrin, 2016):**  booster - gbtree, max depth - 6, learning rate - 0.3, number of estimators - 1000, max depth - 6, reg alpha - 0

- **LightGBM (Ke et al., 2017):** booster - gbdt, max depth - None, learning rate - 0.1, number of estimators - 1000, min data in leaf - 20

- **Random Forests (RF) (Breiman, 2001):**  number of estimators - 1000, criterion - gini, max depth - None, warm start - False

- **Multi-layer Perceptron (MLP):**  Number of layers - 4, hidden units - [feature dimensions, feature dimensions/2, feature dimensions/4, feature dimensions/8], activation function - ReLU, early stopping - True with patience 10, batch size - 256, maximum number of epochs - 200, optimizer - Adam

- **Ridge Regression:** alpha - 1

- **Regression DT:** max depth - 3, criterion - gini

- **Logistic Regression:**  solver - lbfgs, no regularization

- **Classification DT:** max depth - 3, criterion - gini

We follow the default settings for the other hyper-parameters that are not mentioned here.

# B    Performance metrics

- **Mean Absolute Error (MAE):**

$$\text{MAE} = \mathbb{E}_{(\mathbf{x}^t, y^t) \sim \mathcal{P}} ||g^*_{\theta(\mathbf{x}^t)}(\mathbf{x}^t) - y^t)||_1 \simeq \frac{1}{L} \sum_{k=1}^{L} ||g^*_{\theta(\mathbf{x}^t_k)}(\mathbf{x}^t_k) - y^t_k||_1,$$

- **Local MAE (LMAE):**

$$\text{LMAE} = \mathbb{E}_{\mathbf{x}^t \sim \mathcal{P}_X} ||g^*_{\theta(\mathbf{x}^t)}(\mathbf{x}^t) - f^*(\mathbf{x}^t)||_1 \simeq \frac{1}{L} \sum_{k=1}^{L} ||g^*_{\theta(\mathbf{x}^t_k)}(\mathbf{x}^t_k) - f^*(\mathbf{x}^t_k))||_1,$$

- **NSE** (Nash & Sutcliffe, 1970):

$$NSE = 1 - \frac{\mathbb{E}_{\mathbf{x}^t \sim \mathcal{P}_X} ||f^*(\mathbf{x}^t) - g^*_{\theta(\mathbf{x}^t)}(\mathbf{x}^t)||_2^2}{\mathbb{E}_{\mathbf{x}^t \sim \mathcal{P}_X} ||f^*(\mathbf{x}^t) - \mathbb{E}_{\hat{\mathbf{x}}^t \sim \mathcal{P}_X}[f^*(\hat{\mathbf{x}}^t)]||_2^2} \simeq 1 - \frac{\frac{1}{L} \sum_{k=1}^{L} ||f^*(\mathbf{x}^t_k) - g^*_{\theta(\mathbf{x}^t_k)}(\mathbf{x}^t_k)||_2^2}{\frac{1}{L} \sum_{k=1}^{L} ||f^*(\mathbf{x}^t_k) - \frac{1}{L} \sum_{k=1}^{L}[f^*(\mathbf{x}^t_k)]||_2^2}.$$

If $NSE = 1$, the predictions of the locally interpretable model perfectly match the predictions of the black-box model. On the other hand, if $NSE = 0$, the locally interpretable model performs as similar as the constant mean value estimator. If $NSE < 0$, the locally interpretable model performs worse than the constant mean value estimator.

## C  Implementations of benchmark models

In this paper, we use 3 different benchmark models. Implementations of those models can be found in the below links.

- LIME: https://github.com/marcotcr/lime (Ribeiro et al., 2016)

- SILO: https://github.com/GDPlumb/MAPLE (Bloniarz et al., 2016)

- MAPLE: https://github.com/GDPlumb/MAPLE (Plumb et al., 2018)

## D  Data statistics

Table 5: Data Statistics of 5 real-world datasets. Label distributions: Number of positive labels (positive label ratio) for classification problem, and label mean (5%-50%-95% percentiles) for regression problem.

| Problem | Data name | Number of samples | Dimensions | Label distribution |
|---|---|---|---|---|
| Regression | Blog Feedback | 60,021 | 280 | 6.6 (0-0-22) |
| | Facebook Comment | 603,713 | 54 | 7.2 (0-0-30) |
| | News Popularity | 39,644 | 59 | 3395.4 (584-1400-10800) |
| Classification | Adult Income | 48,842 | 108 | 11,687 (23.9%) |
| | Weather | 112,925 | 61 | 25,019 (22.2%) |

## E  Learning curves of LIMIS

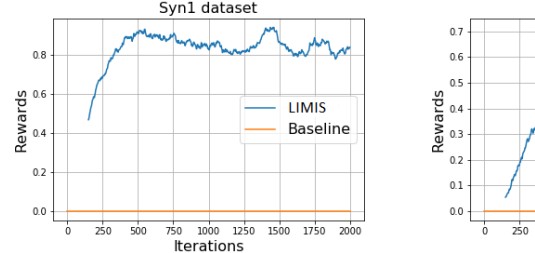 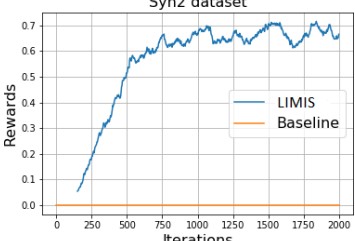 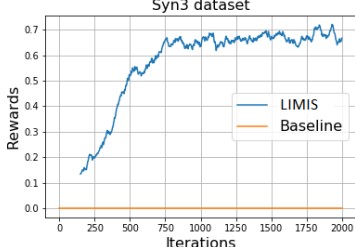

Figure 6: Learning curves of LIMIS on three synthetic datasets. X-axis: The number of iterations on instance-wise weight estimator training, Y-axis: Rewards (LMAE of baseline (globally interpretable model) - LMAE of LIMIS), higher is better.

## F  Instance-wise weight distributions for synthetic datasets

Fig. 7 (a)-(c) show that the instance-wise weights have quite skewed distribution. Some samples (e.g. with average instance-wise weights above 0.5) are much more critical to interpreting the probe sample than many others (e.g. average instance-wise weights below 0.1).

Furthermore, we analyze the instance-wise weights of training samples, and Fig. 8 shows that the training samples near the probe sample get higher weights – LIMIS learns the meaningful distance metrics to measure the relevance while interpreting the probe samples.

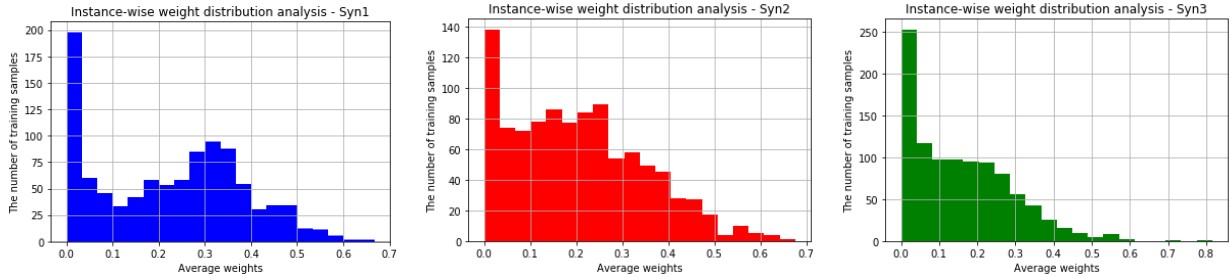

Figure 7: Instance-wise weight distributions for (a) Syn1, (b) Syn2, and (c) Syn3 datasets.

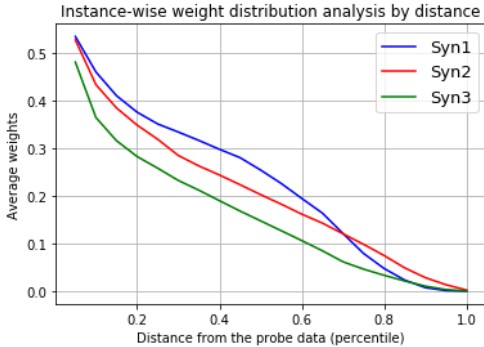

Figure 8: Average instance-wise weights vs. distance from the probe sample.

# G  Additional results

## G.1  Sample complexity analyses with differentiable baselines

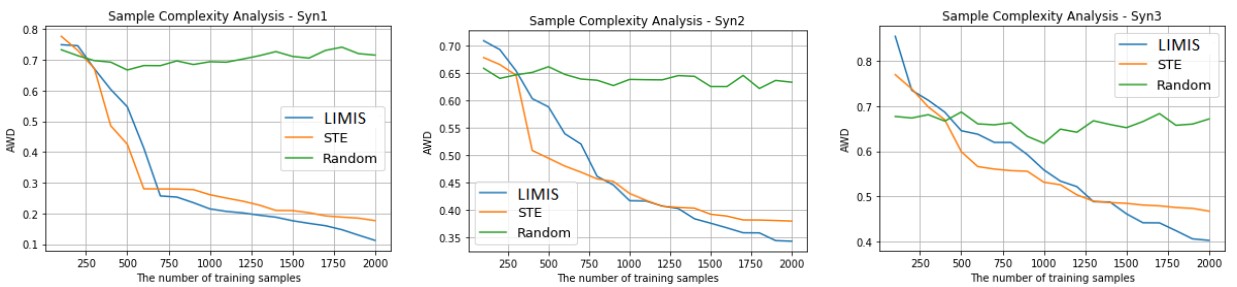

Figure 9: AWD performances in terms of the number of training samples used to train three models: LIMIS, STE and Random.

## G.2  Which training samples are selected by LIMIS, MAPLE and LIME?

LIMIS, MAPLE and LIME select a subset of training samples to construct locally-interpretable models. The training samples selected by LIME are the ones closest to the point to explain. MAPLE utilizes random forest model (trained to predict black-box model outputs) to select the subset of training samples. In this subsection, we quantitatively analyze which samples are chosen by LIMIS, MAPLE and LIME.

Due to the lack of ground truth for ideal training sample selection in real-world datasets, we use synthetic datasets to demonstrate this experiment. Note that for each synthetic data, ideal training sample selections are explicitly determined by $X_{10}$ and $X_{11}$ (see the definitions of Syn1 to Syn3). Therefore, we can quantitatively

evaluate the performances in terms of AUC comparing between selected training samples and ideal training sample selection.

Table 6: Evaluation on correctly selected training samples by LIMIS, LIME, and MAPLE in terms of AUC. **Bold** represents the best.

| Models / Datasets | Syn1 | Syn2 | Syn3 |
|:---:|:---:|:---:|:---:|
| LIMIS | **0.7837** | **0.6892** | **0.6935** |
| LIME | 0.5253 | 0.5017 | 0.5202 |
| MAPLE | 0.6723 | 0.5844 | 0.5452 |

As can be seen in Table 6, the average performance of correctly chosen samples on Syn1 to 3 are 0.7218, 0.5157, 0.6006 using LIMIS, LIME, and MAPLE, indicating the superiority of LIMIS.

### G.3 Additional ablation study - Optimization

To better motivate our method, we perform ablation studies, demonstrating that the proposed complex objective can be efficiently addressed with policy-gradient based RL where the gradient has a closed-form expression. The inner optimization is used for fitting the surrogate explainable model. We explain that for simple surrogate models such as ridge regression, the fitting has a closed form expression and the overall computational complexity is negligible indeed, yielding similar training time compared to the alternative methods. Note that policy-gradient is only utilized for the outer-optimization.

Table 7: Average Weight Difference (AWD) comparisons on three synthetic datasets with different number of train samples (N). Training time is computed on a single K80 GPU until the model convergence (i.e., no more validation fidelity improvements).

| Optimization | Training samples | $N = 1000$ | | $N = 2000$ | |
|:---:|:---:|:---:|:---:|:---:|:---:|
| | | Average performance | Training time | Average performance | Training time |
| **Bi-level** | **LIMIS** | **0.3982** | 49 mins | **0.2936** | 92 mins |
| Single-level | Gumbel-softmax | 0.4209 | 38 mins | 0.3190 | 71 mins |
| | STE | 0.4156 | 39 mins | 0.3208 | 73 mins |
| Two-stage single-level | LIME | 1.6372 | 17 mins | 1.5633 | 21 mins |
| | SILO | 0.6983 | 30 mins | 0.6561 | 44 mins |
| | MAPLE | 0.6217 | 55 mins | 0.5890 | 104 mins |

Table 12 compares our proposed method LIMIS to other methods (Gumbel-softmax and STE) which utilize single-level optimization (i.e. direct back-propagation). LIMIS with bi-level optimization achieves better performance (lower AWD) with small increase in computational complexity. In addition, compared to other baselines (LIME, SILO, and MAPLE) which utilize two-stage optimization (where each stage is single-level), the proposed bi-level optimization in LIMIS shows significantly better performance with similar complexity.

### G.4 Regression with shallow regression DT as the locally interpretable model

Table 8: Overall prediction performance (metric: MAE, lower is better) and fidelity (metric: NSE, higher is better) on real-world regression datasets, using shallow Regression DT as the locally interpretable model while explaining the black box models: XGBoost, LightGBM, MLP and RF. 'Original' represents the performance of the original black-box model, that the locally-interpretable modeling is applied on. We also show the performance of shallow regression RDT as a globally-interpretable model (reported the performance (in terms of MAE) under the data name). Red represents performance that is worse than globally-interpretable shallow regression DT and the negative NSE. **Bold** represents the best results.

| **Datasets** ‖ | Models ‖ | XGBoost | | LightGBM | | MLP | | RF | |
|---|---|---|---|---|---|---|---|---|---|
| (RDT) ‖ | Metrics ‖ | MAE | NSE | MAE | NSE | MAE | NSE | MAE | NSE |
| **Blog** (5.955) | Original | 5.131 | 1.0 | 4.965 | 1.0 | 4.939 | 1.0 | 5.203 | 1.0 |
| | **LIMIS** | **5.121** | **.8242** | **4.778** | **.8939** | **4.587** | **.6375** | **4.652** | **.8990** |
| | LIME | 11.80 | .2658 | 13.22 | .1483 | 7.396 | -.6201 | 19.61 | -.4116 |
| | SILO | 5.149 | .8035 | 4.818 | .8816 | 4.649 | .6177 | 4.715 | .8774 |
| | MAPLE | 5.329 | .7991 | 5.024 | .8660 | 4.609 | .6339 | 5.016 | .8201 |
| **Facebook** (22.28) | Original | 24.18 | 1.0 | 20.22 | 1.0 | 18.36 | 1.0 | 30.09 | 1.0 |
| | **LIMIS** | **21.82** | **.9307** | **21.35** | **.9194** | **18.56** | **.8832** | 22.44 | **.7236** |
| | LIME | 36.69 | .3278 | 44.21 | .1809 | 40.85 | -.1513 | 51.70 | .2301 |
| | SILO | 22.42 | .8655 | 22.33 | .7235 | 19.57 | .8566 | 24.41 | .6917 |
| | MAPLE | 22.15 | .8824 | 23.43 | .8581 | 20.32 | .8035 | 27.12 | .3134 |
| **News** (3093) | Original | 2995 | 1.0 | 3140 | 1.0 | 2255 | 1.0 | 3378 | 1.0 |
| | **LIMIS** | 2938 | **.9382** | **2504** | **.4104** | **2226** | **.9016** | **2431** | **.2768** |
| | LIME | 6272 | -.6267 | 7737 | -2.960 | 2390 | .0013 | 9637 | -7.075 |
| | SILO | **2910** | .1020 | 2854 | .3461 | 2274 | .8201 | 2874 | .2278 |
| | MAPLE | 2968 | .9288 | 2846 | .3631 | 2284 | .8021 | 2888 | .1872 |

Table 9: Fidelity results (metric: LMAE, lower is better) on regression problems with shallow regression DT as the locally interpretable model. **Bold** represents the best results.

| Datasets | Models | XGBoost | LightGBM | MLP | RF |
|---|---|---|---|---|---|
| Blog | **LIMIS** | **.7530** | **1.358** | 1.273 | **1.413** |
| | LIME | 9.160 | 11.16 | 5.006 | 17.461 |
| | SILO | .8325 | 1.379 | **1.178** | 1.934 |
| | MAPLE | 1.029 | 1.598 | 1.359 | 2.158 |
| Facebook | **LIMIS** | **7.240** | **6.867** | **5.596** | **15.77** |
| | LIME | 31.52 | 37.75 | 30.58 | 45.58 |
| | SILO | 8.459 | 9.149 | 6.997 | 18.63 |
| | MAPLE | 7.985 | 8.644 | 7.290 | 23.17 |
| News | **LIMIS** | **389.0** | **1072** | **116.6** | **957.1** |
| | LIME | 4455 | 6243 | 504.0 | 9969 |
| | SILO | 496.7 | 1214 | 160.6 | 1175 |
| | MAPLE | 440.7 | 1201 | 163.6 | 1196 |

### G.5   Regression with RR as the locally interpretable model - Fidelity analysis in Local MAE

Table 10: Fidelity results (metric: LMAE, lower is better) on regression problems with ridge regression as the locally interpretable model. **Bold** represents the best results.

| Datasets | Models | XGBoost | LightGBM | MLP | RF |
|---|---|---|---|---|---|
| Blog | **LIMIS** | **.8679** | **1.135** | **1.432** | **1.651** |
|  | LIME | 6.534 | 8.037 | 8.207 | 17.01 |
|  | SILO | 2.220 | 3.046 | 2.393 | 3.909 |
|  | MAPLE | .9690 | 1.416 | 1.550 | 1.984 |
| Facebook | **LIMIS** | **6.394** | **21.29** | **8.217** | **33.64** |
|  | LIME | 32.57 | 33.70 | 27.38 | 48.03 |
|  | SILO | 19.51 | 30.07 | 11.52 | 40.14 |
|  | MAPLE | 7.664 | 31.25 | 13.31 | 44.38 |
| News | **LIMIS** | **436.9** | **1049** | **74.11** | **905.8** |
|  | LIME | 3317 | 4766 | 327.4 | 8828 |
|  | SILO | 657.2 | 1253 | 79.85 | 1345 |
|  | MAPLE | 500.5 | 1261 | 88.19 | 1157 |

### G.6 Classification with RR as the locally interpretable model

Table 11: Overall prediction performance (metric: APR, higher is better) and fidelity (metric: NSE, higher is better) on real-world classification datasets, using RR as the locally interpretable model while explaining the black box models: XGBoost, LightGBM, MLP and RF. 'Original' represents the performance of the original black-box model, that the locally-interpretable modeling is applied on. We also show the performance of Logistic Regression (LR) as a globally-interpretable model (reported the performance (in terms of APR) under the data name). Red represents performance that is worse than globally-interpretable model logistic regression and the negative NSE. **Bold** represents the best results.

| Datasets | Models | XGBoost | | LightGBM | | MLP | | RF | |
|---|---|---|---|---|---|---|---|---|---|
| | Metrics | APR | NSE | APR | NSE | APR | NSE | APR | NSE |
| **Adult** (.7553) | Original | .8096 | 1.0 | .8254 | 1.0 | .7678 | 1.0 | .7621 | 1.0 |
| | **LIMIS** | **.7977** | **.9871** | **.8039** | **.9439** | .7670 | **.9791** | **.7977** | **.9217** |
| | LIME | .6803 | .7195 | .6805 | .6259 | .6957 | .8310 | .7057 | .6759 |
| | SILO | .7912 | .9750 | .7884 | .9301 | .7655 | .9778 | .7664 | .9140 |
| | MAPLE | .7947 | .9840 | .8011 | .9386 | **.7683** | .9636 | .7958 | .8961 |
| **Weather** (.7009) | Original | .7133 | 1.0 | .7299 | 1.0 | .7205 | 1.0 | .7274 | 1.0 |
| | **LIMIS** | **.7140** | .9879 | **.7290** | **.9801** | .7212 | .9755 | **.7331** | **.9450** |
| | LIME | .6376 | .7898 | .6392 | .6873 | .6395 | .5321 | .6387 | .4513 |
| | SILO | .7134 | .9888 | .7281 | .9773 | **.7220** | **.9797** | .7277 | .9024 |
| | MAPLE | .7134 | **.9897** | .7273 | .9778 | .7213 | .9702 | .7308 | .9323 |

### G.7 Qualitative analysis: LIMIS interpretation

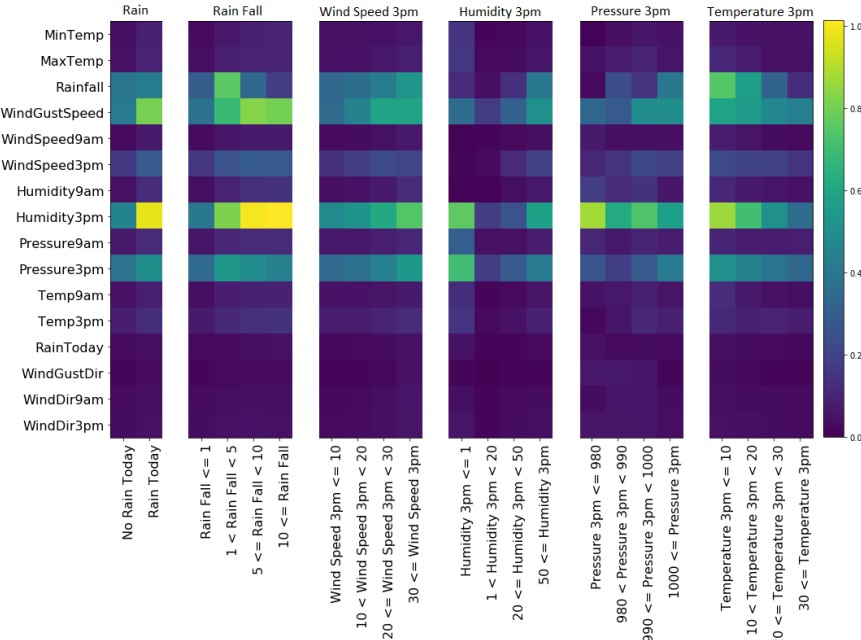

Figure 10: Discovered feature importance (denoted with the colors) by LIMIS on Weather dataset for 6 types of subgroups: (1) Rain, (2) Rain fall, (3) Wind speed 3pm, (4) Humidity 3pm, (5) Pressure 3pm, (6) Temperature 3 pm.

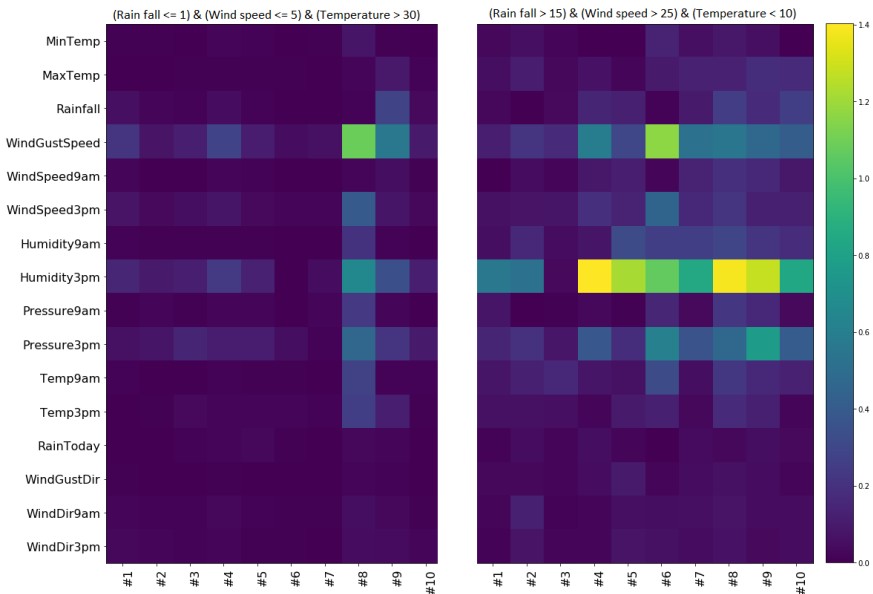

Figure 11: Discovered feature importance on Weather data for: *rain fall ≤ 1, wind speed (at 3pm) ≤ 5, and temperature (at 3pm) > 30* (left), and '*rain fall > 15, wind speed (at 3pm) > 25, and temperature (3pm) < 10* (right).

In this section, we qualitatively analyze the explanations provided by LIMIS. Although LIMIS can provide local explanations for each instance separately, we consider the explanations in subgroup granularity for better visualization and understanding. On Weather dataset, Fig. 10 shows the feature importance (discovered by LIMIS) for six subgroups in predicting whether it will rain tomorrow, using XGBoost as the black-box model. We use RR as the locally interpretable model and the absolute value of fitted coefficients are used as the estimated feature importance. For rain fall subgroups, humidity and wind gust speed seem more important for heavy rain (rain fall ≥ 5) than light rain (rain fall < 5). For temperature subgroups, rainfall, wind gust speed and humidity are more important for cold days (temperature (at 3pm) < 10) than warm day (temperature (at 3pm) ≥ 20). In general, for *heavy rain, fast wind speed, low pressure, and low temperature* subgroups, humidity, wind gust speed and rain fall variables are more important for prediction. Fig. 11 shows the feature importance (discovered by LIMIS) for two subgroups. We observe the clear difference of the impact of afternoon humidity and wind gust speed, on instances that clearly reflect different climate characteristics. This underlines how LIMIS can shed light on the samples with distinct characteristics. Additional use cases for human-in-the-loop AI capabilities can be found in the Sect. 6.

# H   Additional Analyses

## H.1   Synthetic data experiments with LIME

In Fig. 3, we exclude the performance of LIME baseline due to highlighting the performance improvements from the best alternative methods. In Fig. 12, we include the results with LIME baselines for completeness of the synthetic data experiments.

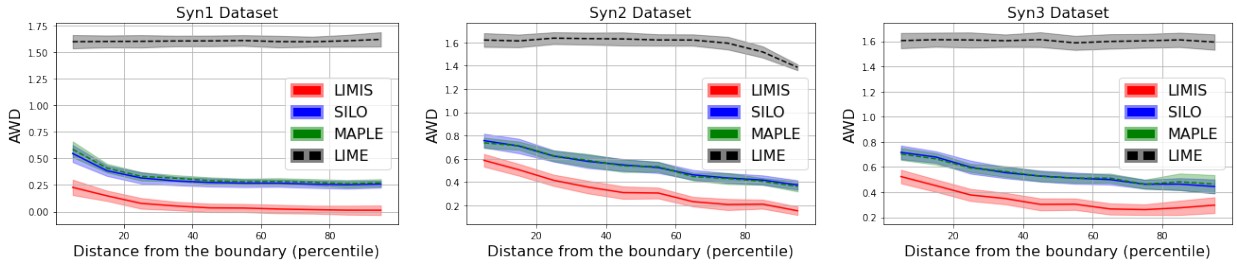

Figure 12: Mean AWD (aggregated per uniformly divided x-axis bin) with 95% confidence intervals (of 10 independent runs) on three synthetic datasets (y-axis) vs. the percentile distance from the boundary where the local function behavior change (x-axis) with LIME as an additional baseline.

## H.2   Scatter plots for synthetic data experiments

In Fig. 3, we report the average AWD performances after aggregating the AWD values per uniformly divided x-axis bins. In this subsection, we report the scatter plots between distance from the boundary (x-axis) and LIMIS's AWD for each sample (y-axis) across 3 different synthetic datasets.

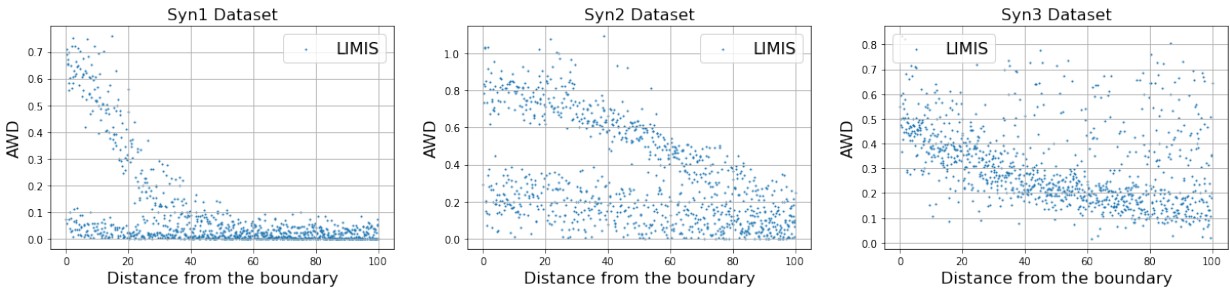

Figure 13: Scatter plots of LIMIS's AWD across distance from the boundary in terms of percentiles.

## H.3   Additional ablation study: Sampling at inference

As explained in Sect. 3, we only use the sampling at training time to encourage exploration. At inference time, we use the weighted optimization using the outputs of the instance-wise weight estimator. In this subsection, we analyze the impact of weighted optimization in comparison to the sampling at inference procedure.

We use XGBoost as the black-box model and RR as the locally interpretable models. As can be seen in Table. 12, sampling at inference consistently shows worse performances than weighted optimization (that is proposed in the LIMIS framework), but overall the differences are small.

## H.4   Local generalization of explanations: Additional datasets

In Table. 3, we provide the results of local generalization of explanations via Neighbor LMAE with Blog and Facebook datasets. In Table. 13, we also provide the Neighbor LMAE results for other 3 real-world datasets: News, Adult, and Weather.

Table 12: Additional ablation studies on 5 real-world datasets with weighted optimization vs. sampling at inference. Metrics are LMAE for both regression and classification datasets.

| Datasets (Metrics: LMAE) | Regression datasets | | | Classification datasets | |
|---|---|---|---|---|---|
| | Blog | Facebook | News | Adult | Weather |
| Weighted optimization at inference | .8679 | 6.394 | 436.9 | .1397 | .1129 |
| Sampling at inference | .8821 | 6.671 | 455.2 | .1465 | .1157 |
| Difference (%) | 1.6% | 4.3% | 4.2% | 4.9% | 2.5% |

Table 13: Prediction performance (metric: *neighborhood* LMAE and *pointwise* LMAE, lower is better) on regression/classification datasets using RR as the locally interpretable model while explaining the black box models: XGBoost (Chen & Guestrin, 2016) and LightGBM (Ke et al., 2017).

| **Datasets** | Models | XGBoost | | | LightGBM | | |
|---|---|---|---|---|---|---|---|
| (RR) | Metrics | *Neighbor* LMAE | *Pointwise* LMAE | Diff | *Neighbor* LMAE | *Pointwise* LMAE | Diff |
| **News** | LIMIS | 442.9 | 436.9 | 1.38% | 1064 | 1049 | 1.47% |
| | LIME | 3550 | 3317 | 7.05% | 5161 | 4766 | 8.30% |
| | SILO | 667.5 | 657.2 | 1.57% | 1258 | 1253 | 0.4% |
| | MAPLE | 501.5 | 500.5 | 0.19% | 1310 | 1261 | 3.9% |
| **Adult** | LIMIS | .1402 | .1397 | 0.39% | .1297 | .1288 | 0.76% |
| | LIME | .3162 | .3117 | 1.46% | .3152 | .2975 | 5.96% |
| | SILO | .1664 | .1622 | 2.59% | .1474 | .1432 | 2.95% |
| | MAPLE | .1717 | .1599 | 7.42% | .1428 | .1407 | 1.50% |
| **Weather** | LIMIS | .1216 | .1129 | 7.7% | .1357 | .1291 | 5.14% |
| | LIME | .3093 | .2750 | 12.5% | .3037 | .2933 | 3.55% |
| | SILO | .1259 | .1257 | 0.21% | .1410 | .1388 | 1.61% |
| | MAPLE | .1294 | .1232 | 5.10% | .1437 | .1371 | 4.39% |

## H.5   Training / Inference time for real-world datasets

In this subsection, we demonstrate the training and inference times of LIMIS on 5 real-world datasets. For training time, we exclude Stage 1, black-box model training, to solely focus on LIMIS-specific instance-wise weight estimator training time. We use a single NVIDIA V100 GPU to train and infer the LIMIS framework.

Table 14: Runtime analyses on 5 real-world datasets. Inference time is computed per one testing sample. Training time is computed until the model convergence (i.e., no more validation fidelity improvements)

| Datasets | Regression datasets | | | Classification datasets | |
|---|---|---|---|---|---|
| | Blog | Facebook | News | Adult | Weather |
| Number of samples | 60,021 | 603,713 | 39,644 | 48,842 | 112,925 |
| Dimensions | 380 | 54 | 59 | 108 | 61 |
| Training time | 56 mins | 3 hours 27 mins | 49 mins | 21 mins | 1 hour 17 mins |
| Inference time | 1.7 secs | 1.2 secs | 1.1 secs | 0.8 secs | 0.7 secs |

## H.6   Convergence plots for real-world datasets

Fig. 6 shows the convergence plots for 3 synthetic datasets. In this subsection, we additionally show the convergence plots for 3 real-world datasets (Adult, Weather, and Blog Feedback). In these experiments, we use XGBoost as the black-box model and RR as the locally interpretable model.

Fig. 14 shows that the convergence of LIMIS is stable for the 3 real-world datasets. The convergence is often observed to be around 1000 iterations. Note that for LIMIS training, we use large batch sizes to reduce the noise in the gradients which would be critical for fast and stable convergence (Stooke & Abbeel, 2018).

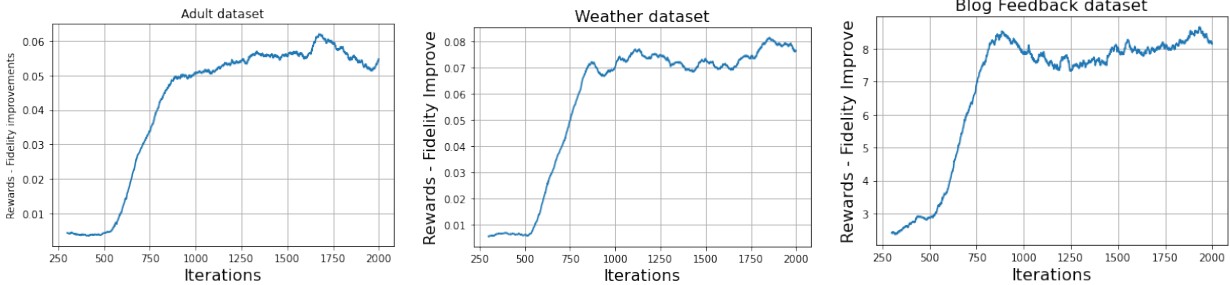

Figure 14: Learning curves of LIMIS on three real-world datasets. X-axis: The number of iterations on instance-wise weight estimator training, Y-axis: Rewards (Fidelity improvements), higher is better.

We also include the convergence plots for STE and Gumbel-softmax variants in Fig. 15. The convergence of the alternatives seems a bit faster but overall the convergence trends seem mostly similar across LIMIS, STE, and Gumbel-softmax.

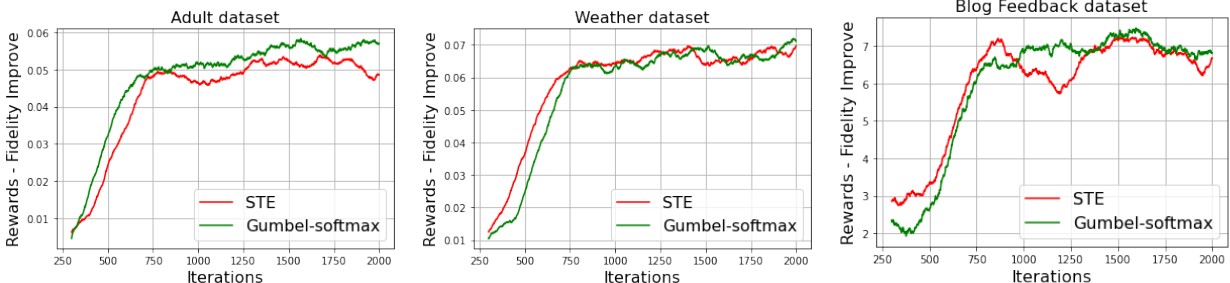

Figure 15: Learning curves of STE and Gumbel-softmax on three real-world datasets. X-axis: The number of iterations on instance-wise weight estimator training, Y-axis: Rewards (Fidelity improvements).

### H.7   Synthetic data with non-linear feature-label relationships

Three synthetic datasets used in the main manuscript have piece-wise linear feature-label relationship. In this subsection, we generalize the results to another synthetic data, with non-linear feature-label relationships. More specifically, we construct Syn4 dataset as follows:

$$Y = f(X) = sin(X_1) + 2cos(X_2) - 0.5(X_3)^2 - exp(-X_4).$$

$X$ are sampled from Uniform$(-1, 1)$ distribution (with 11 dimensions); we set the ground truth local dynamics as the first order coefficients of the Taylor expansion:

$$f'(x) = [cos(x_1), -2sin(x_2), -x_3, exp(-x_4), 0, ..., 0].$$

Then, we utilize LIMIS, LIME, SILO, and MAPLE to recover the ground truth local dynamics with ridge regression as the locally interpretable model. We use the AWD (defined for the first-order coefficients of Taylor expansions as the ground truth explanation) to evaluate the performances of the interpretations. As can be seen in Table 15, the performance of LIMIS is significantly better than other alternatives.

| Datasets / Methods | **LIMIS** | LIME | SILO | MAPLE |
|:---:|:---:|:---:|:---:|:---:|
| Syn4 | **0.2508** | 0.5549 | 0.3411 | 0.3254 |

Table 15: AWD values (the lower, the better) for synthetic data Syn4 whose feature-label relationships are non-linear.

