# OpenReview forum: "LIMIS: Locally Interpretable Modeling using Instance-wise Subsampling"
_TMLR — Accepted by TMLR_

### Review · Reviewer_Hu5A · 2022-07-26

**Summary Of Contributions:**

This paper proposes a new local approximation-based explanation system, LIMIS.

In general, this type of explanation works by sampling some points from a neighborhood around the point being explained and then fitting an interpretable model to those sampled points.  As a result, a key techinical detail is "how are those sampled points chosen?"
-  LIME does this using euclidean distance.
-  SILO/MAPLE do this using a tree ensemble.
-  LIMIS does this using a secondary model that is trained to optimize the fidelity of the resulting explanations.

LIMIS's approach can minimize problems resulting from the difference in the representation capacity between the black-box model being explained and the interpretable model being used as the explanation by:
-  explicitly finding a small set of important points for the explanation to imitate.
-  directly optimizing for fidelity.

Experimentally, the benefits of LIMIS are demonstrated in several ways:
-  Section 4) LIMIS is better able to identify the ground-truth in synthetic experiments.
-  Section 5) LIMIS produces higher fidelity explanations.
-  Section 6) LIMIS is useful in practice (I'm not sure that this section actually supports this).

For TMLR's two reviewing criteria:
-  Does someone care about this problem and these results?  Definitely
-  Are the results and claims supported?  Partially


**Requested Changes:**

Abstract:
-  "An alternative approach is to explain individual predictions using locally interpretable models."
    -  Based on this (and the rest of the abstract), I wasn't certain if the paper is about 1) training a black-box model and then using post-hoc local explanations to explain that model or 2) training a by-design locally explainable model to replace the black-box model entirely.
    -  Please clarify that it is (1).
    -  Minor

Introduction:
-  "They can be utilized ... data sample)."
    -  Please provide for citations that demonstrate these explanations being successfully used for these use cases.
    -  Minor
-  "LIMIS offers the unique capability of instance-based explainability via ranking of the most valuable training
instances"
    -  This claim doesn't seem to be accurate because MAPLE also assigns assigns an importance weight to each training instance based on the instance being explained.
   -  Please revise this claim.
    -  Major

Related Work:
-  "SILO (Bloniarz et al., 2016)) aims to improve LIME."
    -  This doesn't seem to be accurate because 1) SILO doesn't cite or discuss LIME and 2) the author's do not present SILO as an interpretability method (they only mention that local models are interpertable once and do not evaluate SILO from an interpetability perspective).
    -  Please revise this claim.
    -  Major
-  Given Section 2.2 of SILO, how is the use of a tree-ensemble to learn local neighborhood weights "ad-hoc"?
    -  Minor

Synthetic Data Experiments
-  Why should an explanation method that is meant to approximate the black-box model in some neighborhood around the point being explained return exactly "w" near the boundary?  If anything, this seems like it wouldn't let the user know that the black-box model has a discontintuity nearby and, therefore, won't behave as expected.
    -  Please comment on this.
    -  Major
-  For Figure 3, where are LIME's results?
    -  Minor

Robustness of Explantions
-  In the context of explanations, "robustness" is often used to refer to "adversarial robustness" or other measures of how sensitive the explanation is to small perturbations of the point being explained.
    -  Please consider calling this something like "generalization" since this is more aligned with what neighorhood fidelity measures.
    -  Minor
-  What is the neighborhood size used in Table 3?  Without knowing this, these results cannot be interpretted.
    -  Critical
-  Where are the results for the other datasets and black-box models in Table 3 (compared to Table 2)?
    -  Major

LIMIS Explainability Use Cases
-  "we highlight unique explainability capabilities of LIMIS for human-in-the-loop AI deployments."  In what ways are these unique?  LIMIS may produce higher fidelity explanations than LIME/MAPLE, but it isn't clear that LIME/MAPLE can't be used for these use-cases.
    -  Please comment on this.
    -  Major
-  "LIMIS can also be useful for improving a human’s reasoning accuracy, shortening their reasoning time, or increasing their bias detection capability."  This claim does not seem to be supported by the two experiments presented here (which are purely qualitative).
    -  Please comment on this.
    -  Critical
-  "LIMIS does not discover notable biases of the black-box model, via significant dependence on gender, marital status and race features.  These qualitative results demonstrate how the proposed method helps humans to interpret the decision of the machine learning model."  What do direct measurements of bias say? Without knowing whether or not the model is biased, we cannot tell if LIMIS is correct or not.
    -  Please comment on this.
    -  Major
-  "Such a capability can be particularly useful in providing insights on the closest counterfactual input that
would yield a different outcome ... this capability can be efficacious."  Table 4 provides example counterfactuals but doesn't support the claims that this is the *closest* counterfactual or evidence that users can use these counterfactuals to better perform some task.  How does this result show that LIMIS is useful to end-users?
    -  Please comment on this.
    -  Major

**Strengths And Weaknesses:**

Strengths:
-  LIMIS is an intuitive solution to the problem of neighbor selection for local approximation-based explanations.
-  Using synthetic experiments to highlight the failure cases from past methods that LIMIS addresses is compelling.
-  The real dataset experiments show that LIMIS produces higher fidelity explanations than past methods.

Weaknesses:
-  I found "Section 3.1, Stage 2" hard to understand.  Starting with an abstract summary (eg, something like "Our goal is to define x.... to do this we need to define y.... once we have done that, we make simplifications in order to make optimization tractable") and then presenting the details of each of the abstract steps would be helpful (at least for me).
-  I'm not sure that the definition of the ground-truth is as simple as presented in Section 4 (see Requested Changes).  Correctly defining the ground-truth is important for these results to be accurate.
-  I generally find the neighorhood fidelity metric to be much more compelling than the point-wise version (because the latter has pathological failure cases [eg, the useless explanation of $g(x) = 0^t x + f(x)$ gets a perfect score]).  However, the results for this metric have some missing details (see Requested Changes).
-  I'm not sure that the results for Section 6 support the claims (see Requested Changes).  At a high level, it seems like the changes required to support the goal of this section are too large to be made in the review period.  But the paper is publishable without this section (if other concerns are addressed).

---

> ### Author Response · Authors · 2022-08-19
> **Response to the reviewer Hu5A [2/2]**
>
> **Answer 10:** As explained in Figure 3 caption, we exclude LIME due to its poor performance (its AWD is higher than 1.6 in all distance regimes for all datasets which is twice higher than the alternatives). We have added the LIME results in the revised Appendix (Section H.1).
>
> **Answer 11:** We acknowledge this point. To reduce the confusion, we have replaced the term robustness with local generalization in the revised manuscript.
>
> **Answer 12:** Followed by (Plumb et al., 2018), we set sigma = 0.1 as the range of the neighborhood. Note that all the inputs are normalized using standard normalization (with zero mean and unit standard deviation). We have clarified this in the revised manuscript.
>
> **Answer 13:** We understand the importance of the neighborhood LMAE metrics. In the revised appendix, we have added the results with other real-world datasets as well (Section H.4).
>
> **Answer 14:** We acknowledge this point. In the revised manuscript, we have toned down on this part and revised as follows:
> “In this section, we introduce various explainability use-cases of LIMIS for human-in-the-loop AI deployments.” and also added: “As the fidelity of LIMIS is very high, the users would have high trust in the surrogate models.”
>
> **Answer 15:** We have removed this part in the revised manuscript as we did not have experiments to clearly show it.
>
> **Answer 16:** We think the term “bias” makes some confusion here. In the revised manuscript, we have replaced the term bias: “LIMIS can also be used for fairness assessments for black-box models by analyzing the importance of sensitive attribute features locally to see whether any would play a significant role indecision making. For this example, the sensitive attributes (such as gender, marital status and race) are not observed to have high importance for most subgroups.”
>
> **Answer 17:** First, we have added the motivation to Section 6.2 as:
> “In many applications, it is desired to understand what it would take to alter the decision of a model. For example, after rejection of a loan application, the applications would want to understand what they can change to get it accepted, or after a disease diagnosis, doctors and patients would want to understand the suggestions on what can be changed about the patients health to reverse the diagnosis prediction.”
>
> Then, to clarify how Table 4 shows the claims, we have added:
> “In most cases, we observe that the suggestions are consistent with the expectations. For example, better investment outcomes, higher paying jobs and additional education are among common suggestions. When the inputs are modified with these changes, the black-box model predictions change from <50K to >50K, for all cases exemplified here, underlining the accuracy of the suggestions.”
>
> Regarding the closeness of counterfactuals, we have clarified how we identify the samples as :
> “Essentially, our approach to find the modification suggestion relies on traversing the DT upwards from the leaf, and finding the nearest (in terms of being closest in edge distance) node condition that would yield the opposite prediction.”
>
> Although the similarity is not rigorous in a metric space, it should still yield a “perceptually-nearby” sample, that could be useful in real world applications as explained above.

---

> ### Author Response · Authors · 2022-08-19
> **Response to the reviewer Hu5A [1/2]**
>
> Thank you for providing a great summary of our work, positive feedback on its capabilities and performance, and your constructive comments. Hopefully, our responses will help resolve your concerns and questions.
>
> **Answer 1:** Thanks for raising this concern. Indeed, that section contains some key technical contributions so we have substantially modified it to make it more understandable. We have clearly provided the objectives and constraints, and described how to make the optimization problem tractable with the proposed approximations in the revised manuscript.
>
> **Answer 2:** As mentioned in footnote 3 in Section 4, for synthetic data experiments, we use the ground truth data generation functions as the black-box model to interpret instead of training a new black-box model on top of samples generated by the data generation functions. Therefore, the ground-truth definitions are identical as the ground-truth data generation functions. We have clarified this in the revised manuscript.
>
> **Answer 3:** We agree that neighborhood fidelity has clear value in evaluating the robustness of the locally interpretable models, given that the objective of a locally interpretable model is to interpret the black-box model’s local dynamics (not only the testing sample, but also its neighbors). One useful capability of LIMIS is counterfactual modeling (see Section 6.2) and for this, it is critical that the local model is meaningful around the data sample for analysis of output changes given input modifications. We have added clarifications on this metric and hopefully it addresses your questions.
>
> **Answer 4:** The main objective of Section 6 of this manuscript is to show the use-cases of locally interpretable models. In the revised manuscript, we have added more details on the experimental settings, interpretations of the results and discussions in Section 6. We prefer to keep it as it can help most readers to understand the real-world use cases more concretely.
>
> **Answer 5:** We have added some references that utilize the locally interpretable models in various applications.
>
> **Answer 6:** We acknowledge this point. In the revised manuscript, we have modified this claim as follows: “LIMIS can offer the instance-based explainability via ranking of the most valuable training instances.
>
> **Answer 7:** We acknowledge this point. SILO and LIME were concurrent works for locally interpretable modeling. Even though the authors of the SILO paper did not mention interpretability, SILO can be directly utilized as the locally interpretable models as LIME did. In the revised manuscript, we have revised that sentence as: “SILO (Bloniarz et al., 2016) proposed a nonparametric regression based on fitting small-scale local models which can be utilized for locally interpretable models similar to LIME.”
>
> **Answer 8:** SILO first trains a global random forest model using the given training data. This trained random forest model is directly utilized for computing the weights represented in Equation (6). More specifically, weights for each training data is the ratio that the training data is included in the same leaf of the testing sample for the trained tree models. We initially referred to this as “ad-hoc” tree-ensemble model because the tree-ensemble model is trained independent of the locally interpretable models (e.g., learned weights would be the same if we use linear model or shallow decision tree as the locally interpretable model). To reduce the confusion, we have removed the term “ad-hoc” in the revised manuscript.
>
> **Answer 9:** The synthetic datasets used in Section 4 are generated in such a way that we control the ground truth dynamics for quantitative evaluation of explainability outcomes. As discussed in footnote 3, instead of training the black-box model on top of generated samples, we directly use the ground-truth data generation models to generate synthetic data (i.e., Stage 0 and 1 are skipped). The black-box model is identical as the data generation model. Therefore, even near the boundary, we can assume that the ground truth local dynamics are exactly “w”. In the synthetic experiments, we use the linear models as the locally interpretable models which align with the ground truth data generation models (piecewise linear). Therefore, the optimal locally interpretable models should be able to recover the exact ground truth “w”. Note that the “w” is different in different regions. For instance, in Syn1, if X_{10} is negative, w = [1,1,0,0, …]. Otherwise (if X_{10} is positive), w = [0,0,1,1,0,...]. Therefore, the LIMIS can inform whether the black-box model has a discontinuity nearby (around X_{10} = 0) as well.

---

> ### Comment · Reviewer_Hu5A · 2022-08-24
> **Thank you for the clarifications, some of them need more discussion**
>
> The author's revision addressed the majority of my concerns, except for the following:
>
>
> Concern 1:  In the synthetic data, why should the ground truth be "w" near the specified boundary?
> -  The goal of these methods is to explain the model by approximating it over some neighborhood around the point being explained (eg, a normal distribution centered at x with standard deviation 0.1).
> -  As a result, when the point being explained is near the decision boundary, that neighborhood contains points on both sides of the specified boundary and the model's behavior is not necessarily described by either "w".  This is especially true because the model may be discontinuous at the specified boundary.
> -  So it isn't clear to me that either "w" is objectively the "ground truth" for these explanations.
>
> Concern 2:  Use-cases "bias/fairness."
> -  My confusion wasn't about the use of the term bias but rather about the fact that there are tools to measure fairness and yet LIMIS's assessment of fairness is being treated as the ground truth.
> -  For example, the model could be biased and LIMIS could simply be giving a misleading explanation.
> -  As a result, the conclusion based on LIMIS's explanation (that the model isn't biased) should be compared against existing metrics for fairness.
>
> Concern 3:  The general goal of the "use-case" section
> -  Thank you for revising the claims made in this section, they are better supported by the paper now.
> -  However, my concern is that these experiments are still closer to "examples showing that LIMIS could be useful for these applications" than "experiments showing that LIMIS is useful for these things" because the conclusions are based on a few examples.
> -  I'm uncertain as to how major this concern is for TMLR reviewing.

---

> > ### Author Response · Authors · 2022-08-27
> > **Thank you for your careful checking of our responses to the reviews.**
> >
> > Hopefully, the below answers will resolve your additional questions and concerns to our paper.
> >
> > **Answer 1:** We acknowledge your concern for the neighborhood samples around the probe sample having different local dynamics. In our synthetic experiments, each side of the boundary is based on different functions (''w''). Therefore, the neighborhood samples around the probe sample near the boundary can have different local dynamics.
> >
> > The main advantage of the synthetic data experiments is that we can utilize the ground truth local dynamics for the evaluation of locally interpretable models. The ground truth local dynamics are clear for the samples far from the boundary. For the samples near the boundary, as reviewers mentioned, defining the ground truth local dynamics can be tricky (mixture of two local dynamics is not straightforward and may yield infeasible solutions in some situations). Therefore, to make the experiments consistent and robust, we use the ''probe sample'' as the criteria to determine the ground-truth local dynamics. The optimal local explanations for the “probe sample” should still be ''w'' even near the boundary. Note that the same ground truth is applied to compute AWDs for the alternatives; thus, our synthetic experiments are fair for the baselines.
> >
> > One should expect that the accuracy of the local explanations should degrade near the boundary, as the local samples from the other side of the region would start to dominate. Our results also confirm that - LIMIS can recover the ground truth weights (''w'') much better for the samples farther away from the boundary (outperforming others significantly), however, its accuracy gets worse near the transition region. LIMIS is still superior to others even closer to the boundary as it can judiciously identify the most relevant samples for data weighted optimization.
> >
> > **Answer 2:** The main objective of Section 6 is to show some example use-cases of LIMIS, and in this specific case, we aimed to mention that local explanations of LIMIS can provide values for qualitative fairness analyses.
> >
> > We would not want to claim that the LIMIS explanation should be treated as ground truth, for any type of model analyses including fairness analyses. Section 6.1 includes an example with the discovered feature importance (which is a popular approach to explain black-box models) with some qualitative discussions. We did not include quantitative evaluation as the ground-truth feature importance of the black-box model is unavailable.
> >
> > We will tone down the claims to reduce the confusion that the outputs of the LIMIS should be treated as the ground-truth. In addition, we will remove the claim on fairness in Section 6.1 for better clarity.
> >
> > **Answer 3:** Thank you for checking the revised version of Section 6 and acknowledging the improvements.
> >
> > We agree that the results in Section 6 are some ''examples showing that LIMIS could be useful for these applications''. Overall, the main difficulty for real-world data is that there is no ground-truth explanation that we can use to quantitatively evaluate the explanation quality of the LIMIS. Therefore, in Section 6, we focus on showing the possible use-cases of LIMIS in different scenarios, and we tried not to claim that the outputs of the LIMIS are the ground-truth explanations or any quantitative metric for it.
> >
> > We will further clarify this point in the broader impact section.

---

### Review · Reviewer_kxUv · 2022-08-06

**Summary Of Contributions:**

The paper proposes a new method to obtain local interpretation based on instance-wise subsampling.
The method trains a local interpretable model based on instance dependent data-weighted training where weights are from a neural network.
During training, in order to encourage exploration of different weighting, the method selects a subset of data based on sampling,
which causes non-differentiability in the training.
To solve this, the REINFORCE trick is used to train the weight estimation model.
The paper shows that this method significantly outperforms existing methods in terms of explanation quality and prediction accuracy.


**Broader Impact Concerns:**

Although the paper does discuss things like fairness when the Adult dataset is used, this paper doesn't include a section for broader impact or ethical.
Given that the main topic is interpretability, I suggest to add a dedicated section for broader impact on how a more accuract model for local interpretation would help.


**Requested Changes:**

## Must-have
- Explain why data-weighted training would give a model with consistent local interpretation
- More synthetic experiments, especially with different ground-truth models.
- Demonstrate the necessity of using REINFORCE


**Strengths And Weaknesses:**

## Strengths
The method clearly outperforms existing methods on the synthetic datasets in terms of the quality of local explanation.

## Weaknesses
### The choice of data-weighted training for interpretability
It's unclear to me if a model trained using weighted data can be still seen as a model to give the same interpretation.
The paper needs to give some intuitions on why it actually makes sense.
For example, I could imagine changing the weights would dramatically change the decision boundary thus giving a different interpretation.
On this regard, only one experiment with ground-truth interpretation is performed---I would see more to be convinced.
### The necessity of policy gradient
In section 4.3, it argues that using actual sampling with policy gradient outperforms other tricks to deal with discrete sampling for large $N$.
However, the appears to me that the performance of Gumbel-Softmax and STE are pretty close, rendering me the question: Is it worth the extra computation to use the REINFORCE trick here?
Maybe the author could convince me by answering the following questions:
1. How much time does each of the experiment take in the table? Can you also show some training curve to demonstrate how fast each way converges?
2. How the temperature is set for Gumbel-Softmax? Do you use any scheduling on it?
I would suggest the author(s) change(s) the claim if the REINFORCE trick doesn't seem to be necessary.

## Questions
- Where is $N_{mb}$ defined in (2)?
- Would the inconsistency of sampling vs fixed instance-wise weight gives a worse performance? To me it makes more sense to using the targeted number of instances with the highest weights, essentially replacing sampling with the mode.

## Misc
- The notation for multiplication is inconsistent: Sometimes omits the notation means multiplication, sometimes $\cdot$ is used and sometimes $\times$ is used. Please choose one and stick to it.

---

> ### Author Response · Authors · 2022-08-19
> **Response to the reviewer kxUv [2/2]**
>
> **Answer 7:** We have acknowledged it and modified it to use cdot in the revised manuscript in a consistent way.
>
> **Answer 8:** See answer 6.
>
> **Answer 9:** In the main manuscript, we already use three different synthetic datasets to show the outperformance of the proposed method. These three synthetic datasets have different ground-truth models including various non-linearity (polynomial, exponential, etc.) and boundaries. We’d be happy to consider further suggestions if you have any concrete examples that could add value to our paper.
>
> **Answer 10:** Please see answer 2.
>
> **Answer 11:** Thanks for bringing up this important point. We have added the separate Broader Impact section to address the social impacts and limitations of our work.

---

> ### Author Response · Authors · 2022-08-19
> **Response to the reviewer kxUv [1/2]**
>
> Thank you for your valuable comments on our work. Hopefully, our responses will help resolve your questions and concerns.
>
> **Answer 1:** The main objective of this work is to construct an interpretable model that can approximate the local dynamics of black-box models rather than fitting a single global interpretable model to the entire dataset. The locally interpretable models only need to approximate the “local” decision boundary of the black-box model, that would be sample dependent.
>
> Intuitively, some samples would be way more relevant to estimate local decision boundaries for a given sample than some other ones. For example, for the task of income estimation for a young individual, the other young individuals should contain much more relevant information than older individuals, as their incomes would depend on similar factors like the salary of their first jobs rather than the capital gains of their investments like older individuals. Especially given that interpretable models are low capacity, it is crucial to best utilize their capacity by fitting them with the most relevant samples. Thus, weighted optimization is a promising method to minimize the difference between the outputs of the black-box model and locally interpretable model in the local region.
>
> In our experiments, we show that the locally interpretable models can recover the “ground truth local dynamics” not only on a single dataset, but on three different synthetic datasets in Section 4. Also, in Table 3, we show that the proposed method can follow the local dynamics of the black-box models using the neighbor LMAE metric. This indicates that the trained locally interpretable models follow the decision boundary of the black-box models well in the vicinity of samples, as a way of demonstrating the generalization especially for capabilities like counterfactual modeling (Section 6.2), when the goal is to understand how the output changes given the changes in the input.
>
> **Answer 2:** As explained in the manuscript, systematic exploration is critical to explore the large action space (for finding the optimal weights for the training data). Therefore, instead of utilizing end-to-end differentiable models, we proposed an RL based formulation adapting the REINFORCE algorithm that can address the non-differentiability created by the sampling process. We note that we consider two alternative baselines (commonly-used non-differentiable optimization methods) to better showcase the strength of REINFORCE: Gumbel-softmax is an approximation method to convert non-differentiable sampling procedure to the differentiable softmax outputs, while STE replaces the sampling procedure with direct weighted optimization.
>
> As can be seen in Table 1, the average performance improvements with our REINFORCE-based approach is 4.2% and 6.4% for N=1000 and 2000 in comparison with the best alternative. In addition, Figure 9 shows that the improvements increase as the number of samples increases.
>
> Lastly, as can be seen in Table 7, the training time of our REINFORCE-based LIMIS is only 20-25% higher than other differentiable approaches, a very modest increase, while providing very significant, on average 5.3% accuracy improvements.
>
> **Answer 3:** The training time measurements for all real-world experiments are presented in Section H.5 in the revised manuscript. Overall, on the largest dataset (consisting 600k samples), the training time is less than 5 hours with a single GPU. General computational complexity analysis is presented in Section 3.2.
>
> Figure 6 shows the learning curves of LIMIS, demonstrating the model convergence. Within 2,000 iterations, all three learning curves converge. We have also added the convergence curves (Section H.6) for the real-world datasets in the revised manuscript.
>
> **Answer 4:** We fix the temperature to be 0.5. We also tried temperature annealing but it increases the training instability and marginally impact on the performances. Thus, we used a fixed temperature value for the Gumbel-softmax baselines. We have clarified this in the revised manuscript.
>
> **Answer 5:** N_mb is the number of samples in a mini batch. We have clarified it in the revised manuscript.
>
> **Answer 6:** We note that the main objective of the sampling process during training is systematic exploration. At inference time, exploration is no longer needed. Therefore, we use weighted optimization instead of sampling to maximize the inference time performance. We also tried the sampling during the testing time and the performance is consistently worse than the weighted optimization. We have included this ablation study in the revised manuscript (see Section H.3).

---

> > ### Comment · Reviewer_kxUv · 2022-09-03
> > **Reply to author responses**
> >
> > Re. Answer 1: Thanks for the explanation. I'm now happy and believe the the proposed approach is sensible.
> >
> > Re. the REINFORCE trick vs other alternatives (Gumbel-Softmax and STE):
> > - For training time, I'm a bit superised that "our REINFORCE-based LIMIS is only 20-25% higher than other differentiable approaches". Usually REINFORCE would have much larger noise in the gradient thus converges much slower, leading to a noticibly longer training time [1]. How do you compare the training time here? Do they do the same number of iterations or each method stops upon convergences?
> > - Above could be answerd by the convergence curve plots (Figure 6 or Figure 14) but I'm looking for having Gumbel-Softmax and STE on the plot.
> > - Fixing temperature to 0.5 could be the main cause of the slightly worse performance of using Gumbel-Softmax. Can you explain a bit more on what's the training instability when annealing is used?
> >
> > Re. Answer 9: My main concern is that the synthetic dataset(s) used here are all generalized linear models, which are very similar to the local model used. Maybe consider adding a different type of synthetic data.
> >
> > [1] Mohamed, Shakir, et al. "Monte Carlo Gradient Estimation in Machine Learning." J. Mach. Learn. Res. 21.132 (2020): 1-62.

---

> > > ### Author Response · Authors · 2022-09-09
> > > **Second response to the reviewer kxUv**
> > >
> > > Thank you for your reply to our responses to the reviews. Hopefully, the below answers will resolve your additional questions to our manuscript.
> > >
> > > **Answer 1:** We are happy that our answers have resolved your concerns on data-weighted training for local interpretation.
> > >
> > > **Answer 2:** We compute the training time of each method (LIMIS, STE, and Gumbel-softmax) based on training them until convergence, and report the average training time across 10 independent runs. The convergence criteria is the same across different methods - the point that we no longer get validation fidelity improvements.
> > > Note that for LIMIS training, we use large batch sizes to reduce the noise in the gradients, and we observe it to be critical for fast and stable convergence [2]. Our empirical results in Figures 6 and 14 support this - the convergence of LIMIS seems stable across different cases.
> > >
> > > [2] Stooke, Adam, and Pieter Abbeel. "Accelerated methods for deep reinforcement learning." arXiv preprint arXiv:1803.02811 (2018).
> > >
> > > **Answer 3:** We have added the convergence curve plots for STD and Gumbel-Softmax in Figure 15.
> > >
> > > **Answer 4:** We tried multiple different fixed temperature values (0.1, 0.5, 1.0, 2.0, 10.0) for Gumbel-softmax and reported the best performance of Gumbel-softmax (with 0.5 temperature) in the current manuscript.
> > > For temperature annealing, we use the same parameters recommended in the tutorial [Jang, 2022] except the annealing rate (used 0.001) because the total number of iterations is smaller in our experiments. As the temperature decreases, the outputs of the gumbel-softmax would be sparser, which increases the variances of the gradients. That might be the reason for training instability when temperature annealing is applied (we also experienced performance degradation with the fixed temperature of 0.1).
> > > Note that the performances of temperature annealing were worse than the Gumbel-softmax performances with 0.5 fixed temperature. Some previous works also reported that temperature annealing does not help on performance improvements of Gumbel-Softmax [3].
> > >
> > > [3] Maddison, et al. "The Concrete Distribution: A Continuous Relaxation of Discrete Random Variables." ICLR (2017)
> > >
> > > **Answer 5:** The main reason to utilize the synthetic datasets in our manuscript is to quantitatively evaluate the interpretability quality, for which we use AWD (average weight differences) as the quantitative metric. In order to compute AWD, we need (linear) weights as the ground truth interpretations; thus, we use the piecewise-linear functions as the ground truth synthetic data generation models. Note that we use the same locally interpretable model (ridge regression) for the alternatives (LIME, SILO, MAPLE) as well; thus, the comparisons are fair.
> > >
> > > To further verify the superiority of LIMIS, we design another synthetic data with non-linear feature-label relationships. More specifically, we define the Syn4 dataset as follows:
> > >
> > > $Y = f(X) = sin(X_1) + 2 cos(X_2) - 0.5 (X_3)^2 - exp(-X_4).$
> > >
> > > $X$ are sampled from $Uniform(-1, 1)$ distribution (with 11 dimensions); we define the ground truth local dynamics (interpretation) as the first order coefficients of the Taylor expansion:
> > >
> > > $f’(x) = [cos(x_1), -2sin(x_2), -x_3, exp(-x_4), 0, …, 0].$
> > >
> > > Then, we apply LIMIS, LIME, SILO, and MAPLE to recover the ground truth local dynamics with ridge regression as the locally interpretable models. AWD (with first order coefficients of Taylor expansions as the ground truth explanation) is utilized to evaluate the performances of the interpretations of each model. Results show that the performance of LIMIS (AWD: 0.2508) is significantly better (lower is better) than other alternatives (LIME: 0.5549, SILO: 0.3411, MAPLE: 0.3254). We have included the details in Appendix H.7 in the revised manuscript.

---

### Review · Reviewer_ny2W · 2022-08-07

**Summary Of Contributions:**

The paper proposes (and verifies the performance of) a method to train locally interpretable predictors for both classification and regression. The locally interpretable predictor mimics the predictions of highly accurate but not interpretable predictors i.e., black-box predictors. Examples of globally interpretable predictors include a linear model (with linear features) and decision trees of small depth. A locally interpretable predictor’s evaluation at a particular test point is equal to the evaluation of a globally interpretable predictor, but for different test points, the globally interpretable predictors are different from each other.

Methodologically, the paper first postulates that the globally interpretable predictor at each test point is the empirical risk minimizer under a data reweighting; because the data reweighting is different across different test points, the globally interpretable predictors are different across test points, even though the hypothesis class is the same (either linear model or decision trees). The tunable parameters of this locally interpretable predictor is the data reweighting mechanism $h_{\phi}$: the paper uses a policy-gradient routine (Equations 2 and 3) to find a good data reweighting mechanism.

Empirically, the paper shows that the locally interpretable predictor is a high-fidelity approximation of the black-box predictor (Figure 3, Table 1, Table 2, Table 3). They also discuss how the locally interpretable predictor can be used in some explainability use cases: Figure 5 illustrates the discovered feature importance while Figure 6 illustrates counterfactual reasoning.


**Broader Impact Concerns:**

I do not see any ethical concerns arising from this work.

**Requested Changes:**

# Important
Write out all aspects of the optimization problem (objective function, decision variables, constraints etc) in Stage 2.

Explain how Figure 3 was generated (and fix the ticks on the x-axis: if you are reporting percentiles, shouldn’t the ticks range from 0 to 100 rather than from 0 to 1?)

Explain how Figure 5 was generated. Section 6 will be stronger if the paper defines feature importance rigorously and show how the locally interpretable predictor yields feature importance.

# (Less important) discussion questions
Preface: These discussions would be nice to have but not necessary

Does the instance-wise weight estimator provide some form of decision boundary? If it does, can we visualize it? The synthetic datasets by design had decision boundaries; if the instance-wise weight estimator provides an estimate of boundary, can we determine the approximation error?

Is stage 0 and stage 1 necessary? Can we directly use the labeled training data (perhaps with a split into two datasets) to train the instance-wise weight estimator?


**Strengths And Weaknesses:**

# Strengths
In terms of significance, the problem of developing interpretable predictions is important, since people usually need to explain their decision-making process to stakeholders.

In terms of quality, the paper provides good evidence to support that its locally interpretable predictor makes good predictions and provides explanability. I do have questions about the specifics of some of the experiments (please see the Weaknesses section).
- Table 2 shows that the proposed method has higher fidelity than existing baselines to train locally interpretable predictors. It is good to know that baseline techniques on locally interpretable predictions are actually performing worse than the ``sanity check'' of training one globally interpretable model, while the proposed LIMIS is consistently improving upon it.
- The paper provides estimates on computational complexity (Section 3.2) and actual runtimes (Section 5.3), which are useful information to decide whether to run a technique or not.
- It’s good to have the synthetic experiments in which the ground truth is known and can be compared against.
- As somebody new to the local interpretability literature, I greatly appreciate Section 6, which is about explainability use cases.

# Weaknesses
In terms of clarity, the paper is not clear in its discussion of Stage 2, and some experiments are missing important experimental details.
- In the methodological section, the presentation of Equation 1 and the discussion in Stage 2 ultimately talk about different optimization problems. It’s unclear what is the final optimization problem that Stage 2 arrives at after its proposed series of approximation. As a result, I don’t understand to what optimization problem is policy-gradient applied to. To me, Equations 2 and 3 are stochastic gradient descent applied to the $l(\phi)$ objective – I am willing to be convinced otherwise. I also don’t understand the sentence “The sampler block yields a non-differentiable objective …’’, which is tied to the lack of a clearly spelled-out optimization problem.
- Regarding the experiments, I have some questions about experimental details that I could not find in the main text.
  - What do the differentiable baselines in Section 4.3 look like? This is related to the point above about the lack of a clearly spelled-out optimization problem in Stage 2.
  - What does $X_{10}$ mean in the context of Syn1 through Syn3? The data-generating process involves objects like $X_1, X_2$ but there is no explanation of $X_{10}$.
  - How was figure 3 generated? How is the mean AWD computed across 10 runs? Is it taken by averaging the AWD for individual test points, or by aggregating test points whose distance from the boundary lives in a small bin?
    - Relatedly, why is figure 3 a line plot and not a scatter plot? If I understand distance from the boundary and AWD correctly, for each independent run, there should be M (the number of test points) scatter plot points, where the x-axis is the distance of that test point from the boundary, and the y-axis is the AWD for that test point. Perhaps there is some post-processing of the individual scatter plots, since Figure 3 reports results over 10 independent runs – if so, it would be good to discuss.
  - How was Figure 5 generated? There is no discussion of how the feature importance was constructed. Is the feature importance constructed using just the instance-wise weight estimator, or is it constructed using the local predictor as well? What is the exact definition of feature importance?
  - How is the difference between neighbor MAE and pointwise MAE is a measure of robustness? The neighborhood metric is not discussed in either main text or appendix. Since I don’t understand what the neighborhood metric is, I don’t understand how the difference between the two MAEs is a measure of robustness. I am willing to be convinced, but I would need to see the argument clearly spelled out.

---

> ### Author Response · Authors · 2022-08-19
> **Response to the reviewer ny2W [2/2]**
>
> **Answer 8:** For Fig. 5, we use ridge regression as the locally interpretable model. The absolute value of fitted coefficients of the ridge regression model are used as the feature importance estimate. Then, we aggregate (with average) this estimated feature importance across different subgroups and report the aggregated values in Figure 5. We have clarified these in the revised manuscript.
>
> **Answer 9:** Pointwise LMAE is based on computing the differences between black-box predictions and locally interpretable models’ predictions for test samples. To verify that these locally interpretable models can interpret not only the testing samples themselves, but also the neighbors of those testing samples, we compute the neighbor LMAE in addition to pointwise LMAE. More specifically, we randomly sample the neighbors (0.1 standard deviations from the original testing samples) of the testing samples and compute the prediction differences using the black-box and locally interpretable models. If this neighbor LMAE is similar with pointwise LMAE, we can claim that the proposed locally interpretable models are able to interpret the local (neighbor) dynamics, a desired behavior especially for applications like counterfactual modeling (see Section 6.2) that the local model is meaningful around the data sample for analysis of output changes given input modifications. Definition of LMAE can be found in Appendix B.
>
> We have clarified these points in the revised manuscript, hopefully it resolves your concerns.
>
> **Answer 10:** We have substantially modified Stage 2 for better clarity regarding the important aspects of the optimization problem. Our Answers 1 to 4 contain further details.
>
> **Answer 11:** Please see Answer 6-7. We have also fixed the x-axis ticks from 0-1 to 0-100.
>
> **Answer 12:** Please see Answer 8.
>
> **Answer 13:** We can visualize the outputs of the instance-wise weight estimators. Some analyses on the instance-wise weights are already described in Figure 7 and 8. As can be seen in Figure 8, the average instance-wise weights are monotonically decreasing as the distance from the probe data increases. Note that the instance-wise weight estimators are regression models (estimating the optimal continuous weights), it is difficult to visualize the decision boundary like classification models.
>
> **Answer 14:** It depends on the objective of using locally interpretable models. If we want to understand the black-box models that are trained on the given datasets, Stage 0 and Stage 1 are necessary because the main objective is to understand the local dynamics of the black-box models’ decision boundaries. On the other hand, if the goal is to understand the local dynamics of the given datasets, we can skip Stage 0 and 1, and directly run Stage 2 and 3. We have clarified this in the revised manuscript.

---

> ### Author Response · Authors · 2022-08-19
> **Response to the reviewer ny2W [1/2]**
>
> Thank you for providing a well-written summary and highlighting the strength of our paper. Hopefully, our responses will resolve your questions and concerns.
>
> **Answer 1:** Thanks for this important feedback point. Indeed, Stage 2 describes some core methodological innovations so its clarity is essential. We have modified Section 3.1 for stage 2 significantly - we hope the revised manuscript addresses your clarity points. Also, we have added experimental details in the revised manuscript to ensure that the results can be reproducible following the same procedures.
>
> **Answer 2:** Eq. (1) is the main optimization problem that we would like to solve with LIMIS. To make this optimization problem tractable, we apply a series of approximations to the objective and constraints. Followingly, we reformulate Eq. (1) as bi-level optimization problem. Converted tractable optimization problems can be found in Eq. (2) in the revised manuscript.
>
> Eq. (2) (Eq. (3) in the revised manuscript) is a weighted optimization to fit the locally interpretable models. This fitting inside Eq. (2) might be based on gradient descent based optimization for differentiable models like linear regression, or directly using closed-form solution via feature matrix inversion, or model specific methods as information gain based training for decision trees. Note that Eq. (2) comes from the constraints of Eq. (1) after applying aforementioned approximations to the constraints.
>
> In Eq. (3) (Eq. (4) in the revised manuscript), we use the stochastic gradient to optimize the instance-wise weight estimator. Here, l(\phi) is the approximated version of the objective in Eq. (1). We have clarified these points to make Stage 2 clearer in the revised manuscript.
>
> **Answer 3:** As described in the beginning of Stage 2, we mention that we use a probabilistic selection approach with a sampler and use it as the weights for training the locally interpretable models. Sampling is a non-differentiable operation [1, 2]; thus, we explicitly mention that the objective function is also non-differentiable.
>
> [1] Blei, D. M., Kucukelbir, A., & McAuliffe, J. D. (2017). Variational inference: A review for statisticians. Journal of the American Statistical Association, 112(518), 859–877.
> [2] Doersch, C. (2016). Tutorial on variational autoencoders. ArXiv Preprint ArXiv:1606.05908.
>
> **Answer 4:** As mentioned above, LIMIS utilizes a sampling procedure; thus, the objective (loss) function is non-differentiable. In that case, we cannot train the model in an end-to-end way using stochastic gradient descent (SGD).
>
> There are multiple ways to train for non-differentiable objectives in the literature. In LIMIS, we use the REINFORCE algorithm to train the model with non-differentiable objective. We also consider two alternative baselines to better showcase the strength of REINFORCE: Gumbel-softmax is an approximation method to convert non-differentiable sampling procedure to the differentiable softmax outputs, while STE replaces the sampling procedure with direct weighted optimization. Both STE and Gumbel-softmax baselines are differentiable approaches and we can optimize the models in an end-to-end way via SGD.
>
> **Answer 5:** As the second paragraph of Section 4 describes, the synthetic datasets contain the inputs X that are 11-dimensional. Y are directly constructed from these features and do not directly depend on X_{10} and X_{11}. However, X_{10} and X_{11} determine how Y depends on X_1, X_2, X_3, and X_4. For instance, in the Syn1 dataset, Y directly depends on X_1 and X_2 if X_{10} are negative. If X_{10} are positive, Y are dependent on X_3 and X_4 but are independent of X_1 and X_2. We have added this example in the revised manuscript.
>
> **Answer 6:** We first divided the x-axis into 10 uniform bins. Then, we computed each sample’s AWD and aggregated them per each bin. We estimated these average aggregated AWD per bin for 10 runs and reported mean and standard deviations in Figure 3.
>
> **Answer 7:** It is a good point that the raw evaluated AWD values can be more clearly described using scatter plots. To make better visualization, we aggregated those scattered dots into separate bins and reported the mean per each bin. Then, we computed the standard deviations of those aggregated average values and reported them in Figure 3. In the revised appendix (Section H.2), we have also added the scattered plots and clarified how we generate Figure 3 in the caption.

---

> > ### Comment · Reviewer_ny2W · 2022-09-07
> > **Reply to Authors' Responses**
> >
> > Thank you for engaging with my reviews. I am glad to see the changes you have made. I have a small recommendation about Stage 2 and Equation 2 as they are in the current revision. In Equation 2, technically speaking, the parameter $\phi$ does not appear in either the optimization problem's objective or its constraint. For sure, the \emph{distribution} of $c_i(x_j^p)$ depends on $\phi$, but actual realizations of the random variables do not depend on $\phi$. I encourage you to find a way to convey this nuance in the optimization problem.

---

> > > ### Author Response · Authors · 2022-09-09
> > > **Second response to the reviewer ny2W**
> > >
> > > Thank you for your insightful feedback points that have helped us to improve the quality of our manuscript.
> > >
> > > **Answer 1:** Thank you for the suggestion. We agree that $\phi$ is explicitly included in both objective and constraints in Equation (1) but implicitly included in Equation (2). As mentioned, the dependency of $\phi$ in Equation (2) is implicitly included in $c_i(x^p_j)$ and the details are discussed in the text between Equation (1) and (2). In the revised manuscript, we have explicitly added the dependency between $c_i(x^p_j)$ and $\phi$ in Equation (2) as $c_i(\textbf{x}_j^p) \sim Ber(h_\phi (\textbf{x}_j^p, \textbf{x}_i, f^*(\textbf{x}_i)))$.

---

### Review · Reviewer_2fzZ · 2022-08-08

**Summary Of Contributions:**

This paper proposes a new approach to local model interpretability based on an instance-wise subsampling method. At the core of the method -- and its main difference compared to other local interpretability methods -- is that it fits an instance-wise weight estimator to identify the importance of the training samples to explain a test sample. A locally-interpretable model is then trained on this subset of examples, obtaining the explanation. The method is validated on various synthetic and simple ML benchmark datasets, showing strong performance in terms of fidelity/robustness quantitative metrics when compared to other similar interpretability methods.


**Broader Impact Concerns:**


There is no broader impact statement. But it is my opinion that **any paper proposing a interpretability method for ML should come with disclaimers of its limitations**. This is particularly important given the widespread mis-use and mis-understanding of explainable AI methods (see, e.g. Kaur et al "Interpreting Interpretability: Understanding Data Scientists’ Use of Interpretability Tools for Machine Learning", CHI 2020).

**Requested Changes:**


- Provide explanation for the discrepancy between the hard (sampled) weights during training of the instance-wise weight estimation model, compared to the use of soft (non-sampled) weights during test time. [Critical]
- Add a broader impact statement with disclaimers and discussion of its limitations for use in real applications [Critical]
- Validation with human subjects [Nice to Have].
- Remove or re-frame Section 6 as it currently seems to serve to specific purpose [Nice to Have]
- A more detail runtime analysis of training vs testing of the model in the all the benchmark datasets not just Facebook. In particular: effect of lambda and convergence criterion for Algorithm 1. [Critical]
- A deeper qualitative analysis of the instance-wise weight estimator in the benchmark datasets. It would be interesting to see what kind of examples it up/down-weights in which situations. What can we learn about the model from the lens of instance-wise importance? [Nice to Have]
- Typos: "they should" -> "it should" in Ph 2
- In-text citations should use \citet instead of \citep

**Strengths And Weaknesses:**


[Strengths]
- The paper combines ideas from sub-sample selection and local model interpretability in a compelling way
- For a purely technical perspective the approach seems sound and correct. The use of REINFORCE for sampling discrete sample weights seems reasonable (and perhaps unavoidable). My only question/comment in this regard is that it's not clear to me why the training phase samples discrete {0,1} weights for fitting the locally-interpretable models (Algo 1: 5) whereas at test time the dense [0,1] weights are used instead (Algo 2: 3). The reason for this discrepancy does not seem to be discussed in the paper.
- From a purely quantitative standpoint, the method seems to be perform well in terms of local functional approximation compared to other methods
- The paper overall is well written, easy to follow, and clear in its argumentation and derivations

[Weaknesses]
- The method puts a heavy burden of interpretability on the instance-wise weight estimation model, which raises some issues (see below for details). The instance-wise weight estimation model (which is itself a deep learning model) seems key to achieve strong local fidelity but at the same time can potentially transfer the 'black-box-ness' from one  place to the other. In other words, the complex and highly-nonlinear interaction between sample selection of locally-fitted model might be hard to explain to a user, and its effect on the is overall interpretability of the approach is unclear.
- The experiments leave many unanswered questions and could use more explanation, better baselines, and more analysis. In particular:
    - The hand-crafted datasets (the 'black-box models' in this case) seem to be perfect fits for this method: piecewise linear functions over some linear and some non-linear regions. Clearly, any method with any reasonable 'bandwidth'/sample selection and locally-fit linear models should be able to recover the functions. So the results of Section 4.1 don't provide much information, and SILO and MAPLE seem like poor choices of baselines for this task (there's a very extensive literature of locally-linear interpretability models - why not pick one of those?). In addition, the reasons for LIME failing in this task are not very clear to me.
     - In Section 4.2, the paper states "if λ is too small, LIMIS selects too many instances and deteriorates fidelity (due to overfitting)". Is this for a ridge regression local model? How can a linear regression model overfit? Is this Kernel RR?. Furthermore, the results in Fig  4 are, again, not very insightful, except to show that this method has a heavy dependence on the λ parameter, and its impact of the interpretability itself might be strong.
     - The results of 4.3 are perhaps the most informative of this Section, and highlight that that ideal use cases of this method are on large-N scenarios
- I am not sure what's the point of Section 6: LIMIS Explainability Use Cases. It doesn't seem to specifically discuss use cases of this method (especially when compared to other methods). Figure 5 is poorly explained, and its interpretation requires heavy hand-holding. The examples in 6.2 rely on a method to generate counterfactuals (shortest path from prediction to desired outcome) that warrants further motivation. At any rate, these are two experiments where some form of human validation would have been useful. In 2022, an interpretability paper without an attempt to evaluate the method in-context with human subjects does a disservice to itself.

---

> ### Author Response · Authors · 2022-08-19
> **Response to the reviewer 2fzZ [2/2]**
>
> **Answer 5:** We have substantially revised Section 6 to address the issues and hopefully it is clearer now. We have included more details on how we generate Figure 5 and Table 4 in the revised manuscript.
>
> Due to the limited timeframe for the rebuttal period, unfortunately it’s infeasible for us to add human evaluations that would be meaningful (e.g. with very high number of subjects etc.). However, we note that the quantitative evaluations we provide should showcase the high quality explanations. One such metric for this is fidelity on real world data - we show much higher fidelity compared to other methods, including LIME. In addition, we provide some examples in Section 6 on how these can bring value in identifying the characteristics of different data subsets, and accurate counterfactual modeling.
>
> We have also expanded the Broader Impact section to better describe the use cases and the impact LIMIS can bring.
>
> **Answer 6:** As explained in Section 3.2, to encourage exploration during the training, we use probabilistic selection for instance-wise weight estimation models. This would be critical for efficiently exploring large search spaces. At inference time, exploration is no longer needed; it would be better to minimize the randomness to maximize the fidelity of the locally interpretable models. Therefore, we use soft weights at inference time. To quantitatively verify the impact of soft weights at inference time, we have included additional ablation studies to compare soft weights and hard weights performances. As can be seen in the revised appendix (Section H.3), the performances with sampling at inference show consistently worse performances in comparison to the original LIMIS framework with weighted optimization at inference.
>
> **Answer 7:** That’s a great point - we have added a Broader Impact section at the end of the manuscript to discuss the limitations and the social impact of our work.
>
> **Answer 8:** Due to the limited timeframe for the rebuttal period, unfortunately it’s infeasible for us to add human evaluations that would be meaningful (e.g. with very high number of subjects etc.). However, we note that the quantitative evaluations we provide should showcase the high quality explanations. One such metric for this is fidelity on real world data - we show much higher fidelity compared to other methods, including LIME. In addition, we provide some examples in Section 6 on how these can bring value in identifying the characteristics of different data subsets, and accurate counterfactual modeling.
>
> **Answer 9:** We have substantially revised Section 6 to address the issues and hopefully it is clearer now. We have clarified the objectives, methods and takeaways. We hope that the newer version serves its purpose in a better way.
>
> **Answer 10:** In the manuscript, we already include the runtime analyses of training and testing of the models for the synthetic datasets (see Table 7). The main reason that we include the runtime analyses for the Facebook dataset is to highlight scalability, as it is the largest dataset used in the experiments.
>
> As the convergence criteria, we stop training (Algorithm 1) if there are no fidelity improvements. In the revised manuscript and Appendix, we have added the convergence criteria details and runtime analyses (Section H.5) for other benchmark datasets.
>
> **Answer 11:** Thanks for this useful suggestion. Unfortunately, the qualitative example based analyses using real-world tabular datasets are somewhat tricky. Instead, we provide some visualizations of the instance-wise weights in Figure 7 and 8 using the synthetic datasets. As expected, the average instance-wise weights are decreasing as the distance from the probe data is increasing.
>
> **Answer 12:** We have fixed it.
>
> **Answer 13:** We have fixed it.
>
> **Answer 14:** We have also expanded the Broader Impact section to better describe the use cases and the impact LIMIS can bring.

---

> ### Author Response · Authors · 2022-08-19
> **Response to the reviewer 2fzZ [1/2]**
>
> Thank you for recognizing multiple strengths in our work. Below you can find detailed responses to your question and concerns.
>
> **Answer 1:** We acknowledge this important point. The proposed instance-wise weight estimator is indeed a black-box model and difficult to interpret without post-hoc interpretable methods. However, the main value proposition of LIMIS is that the final output, the locally interpretable model, is fully interpretable and the users can utilize the final output for understanding the rationale behind the local decision making process of the black-box model. With our experiments, we show that the fidelity metrics of the locally interpretable models of LIMIS are high, in other words, they approximate the black-box model functionality very well for each sample locally.
>
> Broadly, there are many different forms of explainable AI approaches, from single interpretable models to post-hoc methods for complex black-box models. LIMIS is not an alternative to all of them, but it specifically provides the locally interpretable modeling capability, which has numerous impactful use cases in real world AI deployments, including Finance, Healthcare, Policy, Law, Recruiting, Physical Sciences, and Retail. We have modified the Introduction and Broader impact section to be clearer about this point, hopefully it would resolve the concerns.
>
> **Answer 2:** First and foremost, please note that LIMIS can be applicable to any dataset and interpretable model, not only to piecewise linear cases. For example, LIMIS can be combined with shallow decision trees that have highly nonlinear mapping capability, and can be used to interpret the black-box models with non-piecewise linear cases (please see the results with real-world datasets in Section 5 - Table 2 with two classification datasets).
>
> SILO and MAPLE are two competitive methods to recover the local dynamics of the black-box models. SILO and MAPLE also utilize the linear functions as the locally interpretable models; thus, there is no reason that SILO and MAPLE would fail while LIMIS would be succeeding in the synthetic data experiments - the comparison seems fair in determining the locally interpretable modeling capabilities.
>
> In addition, we would like to add that even though the synthetic datasets are piecewise-linear, the decision boundary is nonlinear and unknown. In that case, even with more complex locally-fit linear models, we cannot guarantee that those models would do better.
>
> Regarding the underperformance of LIME, we attribute it to the fact that LIME utilizes the Euclidean distance to identify the neighbor samples. If only the subsets of the features are dependent on the targets, LIME’s neighbor sampling process would be noisy which makes LIME perform much worse than alternatives (note that MAPLE and SILO directly utilize the target related features for weighting). In general, perceptual relevance of samples may be via highly nonlinear relationships, which is inherently captured in LIMIS via the instance-wise weight estimator, whereas the Euclidean distance based relevance mechanism of LIME would not be able to capture such cases.
>
> **Answer 3:** We acknowledge that the term ‘overfitting’ creates confusion. Overfitting here was broadly considered as the phenomenon of train-test mismatch in performance. For linear regression, overfitting may occur if there are some input features irrelevant to the target, leading the model to focus on spurious correlations while training and thus generalizing to unseen samples poorly. In this example, we are using ridge regression as the locally interpretable model, without any kernel. With a small lambda, LIMIS is not sufficiently encouraged to select the most relevant instances related to the local dynamics to fit the ridge regression model, thus is more prone to learning feature relationships that may generalize poorly. We have revised our expression to be more precise and clearer in Section 4.2.
>
> Figure 4 aims to show the impact of lambda for both subset selection and local fidelity to give insights on how LIMIS works as expected. Judicious subset selection is indeed key to the performance, to best utilize the limited capacity of the interpretable models, so these results are consistent with the message we would like to convey. Note that we optimize lambda value using cross-validation as described in a footnote before, which we have made clearer to prevent concerns.
>
> **Answer 4:** We agree that Section 4.3 highlights some informative results. We note that the ideal use-case of LIMIS is not necessarily large-N scenarios, although real-world datasets are growing fast in number of samples. As described in Table 1, even with N = 1000, a much smaller number of samples than the majority of machine learning use cases, LIMIS outperforms other alternatives.

---

### Author Response · Authors · 2022-08-19
**Thank you for all the valuable comments!**

Thank you very much for all the insightful comments from the reviewers.

We posted the response to each reviewer's comments below. In addition, we also uploaded the revised manuscript.

Hopefully, those responses and the revised manuscript can resolve the reviewers' questions and concerns in this work.

---

### Decision · Action_Editors · 2022-09-14

**Recommendation:** Accept with minor revision

**Comment:**

Thank you to the authors and reviewers for the great discussions.   Throughout the discussions, most of the critical questions/concerns raised by the reviewers have been clarified/addressed.  All reviewers believe the updated version reads better and crosses the threshold for acceptance.   Some of the remaining issues that authors need to consider addressing/discussing include 1) the fairness claim and the general goal of the "use-case" section raised by Reviewer Hu5A, and 2) the issue of the non-interpretability of the weight estimation model and the relatively limited scope of the experiments raised by Reviewer 2fzZ.  Again, thank you very much for the great work of both the reviewers and authors.

Best,

AE

---

> ### Author Response · Authors · 2022-09-21
> **Thank you all - Camera-ready manuscript is submitted.**
>
> Thanks for the editor and all reviewers’ insightful comments. All those thoughtful comments were well addressed in the camera-ready manuscript which is significantly improved from the original submission. Thanks again for the constructive feedback and discussions during the review process.
>
> **Answer 1:** In the camera-ready manuscript, we have toned down the claims to reduce the confusion that the outputs of the LIMIS should be treated as the ground-truth. In addition, we have removed the claim on fairness in Section 6.1 for better clarity.
>
> We have also clarified that the experimental results in Section 6 should be consumed as some examples showing that LIMIS could be useful for these applications. We focused on showing the possible use-cases of LIMIS in different scenarios. We have further clarified this point in the broader impact section of the camera-ready manuscript.
>
> **Answer 2:** In the Broader Impact section of the camera-ready manuscript, we have improved our descriptions on the value proposition of LIMIS, emphasizing that the final output is fully interpretable even though the weight estimation model is not interpretable. Experiments with human subjects would be a great direction for future work to strengthen the proposed framework - we have explicitly included this point in the Broader Impact section of the camera-ready manuscript.